# Consequences of platelet-educated cancer cells on the expression of inflammatory and metastatic glycoproteins

**Mélanie Langiu[1], Lydie Crescence[1,2], Diane Mège[1,3], Christophe Dubois**[1,2]*****,
**Laurence Panicot-Dubois**[1,2]

**1** Aix Marseille University, INRAE 1260 (Institut National de la Recherche Agronomique et de l'Environnement), INSERM 1263 (Institut National de la Santé et de la Recherche), C2VN (Center for CardioVascular and Nutrition Research), Marseille, France, **2** Marseille University, PIVMI (Plateforme d'Imagerie Vasculaire et de Microscopie Intravitale), C2VN (Center for CardioVascular and Nutrition Research), Marseille, France, **3** Department of Digestive Surgery, La Timone University Hospital, Marseille, France

* christophe.dubois@univ-amu.fr

## Abstract

Cancer-associated thrombosis, a major cause of mortality in cancer patients, exhibits a 4 to 7 times higher incidence compared to the general population. Platelet activation by tumor cells contributes to this pro-thrombotic state. Cancer cell-educated platelets have also been described to be implicated in promoting metastasis. Intriguingly, our team, among others, unveils a reverse process, wherein platelets educate cancer cells by transferring lipids, RNAs, and proteins. Here, focusing on colorectal and pancreatic cancers, our study investigates genes and proteins mediating platelet education of cancer cells. We demonstrated, for the first time, that platelets can educate cancer cells by inducing changes in the transcription of genes related to glycosylation, inflammation, and metastasis in cancer cells themselves. These results indicate a direct impact of platelets on cancer cell phenotype. This novel insight suggests potential therapeutic avenues for cancer treatment, disrupting platelet-mediated alterations and influencing the course of cancer progression.

## Introduction

Over a century ago, Armand Trousseau demonstrated a strong link between cancer and thrombosis [1]. Trousseau's Syndrome describes that cancer patients are three times more likely to develop deep vein thrombosis than those without cancer [2]. Consequently, cancer-associated thrombosis (CAT) is the second leading cause of death after the progression of the disease itself [3]. This hypercoagulable state in cancer patients is mainly due to the ability of cancer cells to activate the blood coagulation cascade and platelets by various mechanisms [4]. This lead, not only to the formation of CAT, but may also directly affect the tumor behavior and the formation of metastasis. Different studies have suggested that the use of anti-platelet drugs such as aspirin can prevent cancer from developing and spreading to distant sites [5] We previously revealed, in a mouse model of

**Data availability statement:** All relevant data are within the manuscript and its Supporting Information files.

**Funding:** This work has received support in the form of a grant from the French government under the France 2030 Investment Plan as part of the Initiative of Excellence at Aix-Marseille University [A*MIDEX AMX-22RE-V2-012]. This grant was used to purchase the DuoSpark device used in this study. The funders had no role in study design, data collection and analysis, decision to publish, or preparation of the manuscript.

**Competing interests:** The authors have declared that no competing interests exist.

pancreatic cancer, that Clopidogrel, an antagonist of platelet P2Y12 ADP receptors, limits cancer-associated thrombosis, tumor progression, and metastasis [6]. The activation of platelets and their interaction with cancer cells led to the concept of cancer cells-educated platelets. Several studies have shown that cancer cells-educated platelets facilitate the transport of cancer cells in the bloodstream, their extravasation into tissues, the colonization of distant sites and the growth of metastases [7]. In the tumor microenvironment, the role of extravasated intratumoral platelets is more controversial. On one hand, it is associated with the progression of disease, EMT markers, chemoresistance, immune invasion, and tumor angiogenesis [8–10]. On the other hand, we recently demonstrated that the cadherin-6-dependent interaction between cancer cells and platelets inhibits tumor growth [11]. Interestingly, these studies also highlighted the opposite concept of plated-educated cancer cells. Indeed, platelets can modulate cancer cell behavior through both direct and indirect mechanisms. Direct interactions involve the transfer of platelet biomolecules such as lipids, proteins, and RNA from platelets to cancer cells. In contrast, the indirect pathway occurs when the interaction with platelets induces the activation of signaling pathways in cancer cells themselves, leading to a change in their molecular composition and, ultimately, in their behavior [11,12]. However, to date, the genes, RNA, and proteins involved in this "education" of cancer cells by platelets are not characterized.

Since digestive cancers (including colorectal and pancreatic) are associated with a thrombotic risk, we focused our study on the role of platelet-educated cancer cells in these types of cancers [4]. Specifically, we sought to further characterize what platelets can transfer to cancer cells to change their behavior.

Deregulation of protein glycosylation has recently become a hallmark of cancer, which explains why most proteins involved in metastasis are aberrantly glycosylated [13]. Platelets have been described to be able to release functional glycosyltransferases (GT) as well as associated sugar donors, which allow them to glycosylate extracellular acceptor molecules [14]. Therefore, we focused our studies on the consequences of cancer cell education on proteins involved in glycosylation, inflammation and metastasis. We identified that the interaction of cancer cells with platelets induced a change in the transcription of GT-encoding genes in the cancer cells themselves. We also found that 124 genes encoding for proteins associated with inflammation and metastasis were overexpressed in platelet-educated cancer cells, including fibronectin (FN1), a key glycoprotein involved in the adhesion processes. Taken together, our results demonstrate that platelets influence the composition of cancer cells, which may lead to changes in their behavior within the tumor microenvironment as well as the formation of metastasis.

## Materials and methods

### Cell culture

The mouse pancreatic cancer cell line PANCO2 was generously provided by Dr Ruben Hernandez-Alcoceba (University of Navarra, Pamplona, Spain) to Dr Laurence Panicot-Dubois. Pancreatic cancer cell lines (mouse and human) PANCO2, BxPC-3 (CRL-1687, ATCC), MIA PaCa-2 (CRM-CRL-1420, ATCC), Capan2 (HTB-80, ATCC) and PANC-1 (CRL-1469, ATCC) and the human colorectal cancer cell line HT29 (HTB-38, ATCC) were grown in RPMI-1640 (31870025, Thermofisher Scientific), DMEM (11965092, Thermofisher Scientific), or McCoy's medium (16600082, Thermofisher Scientific) supplemented with 10% of heat-inactivated fetal calf serum, 100 U/ml penicillin (Thermofisher Scientific), 100 µg/ml streptomycin (Thermofisher Scientific), and 2 mM glutamine (Thermofisher Scientific). The human breast cancer cell line MCF-7 (HTB-22, ATTC) was used as a positive control to detect

the expression of GALNT3. All the cells used were tested negative for mycoplasma and grown mycoplasma-free at 37°C in a humidified atmosphere containing 5% $CO_2$.

## Cell transfection

PANCO2 low fibronectin (PANCO2 low FN1) or PANCO2 mock cells were obtained after transfection with pcDNA TM6.2-GW/EmGFP-miR plasmid containing FN1 or non-targeting control siRNA, respectively, using Lipofectamine 2000. Cells carrying the plasmid were then selected for resistance to blasticidin (10 μg/mL in complete medium).

## Preparation of human washed platelets

Blood was obtained from the French Blood Institute "Etablissement Français du Sang (EFS)" (agreement 7580). Healthy volunteers over 18 years signed a written informed consent form before blood sampling. Experiments were performed after obtention of the IRB approval (number 2022-05-12-008) on the 16th of May 2022. Blood was collected in 5 mL citrated tubes. All blood samples were centrifuged at 200 g for 13 minutes at 37°C to obtain plasma rich platelets (PRP). PRP was then centrifuged at 900 g for 13 minutes at 37°C to obtain platelet pellets. Pellets were gently resuspended in CGS buffer (120 mM NaCl, 13 mM trisodium citrate, 30 mM dextrose, pH 7.4) supplemented with 0.02 U/mL apyrase (A6535, Sigma Aldrich) and 0.5 μM PGI2 (538925, Calbiochem). The platelets were washed twice in CGS buffer. Finally, platelets were resuspended in CGS buffer without apyrase and prostacyclin. Platelet concentration was obtained by flow cytometry using an antibody against human CD41 labelled with PC7 (6607115, Beckman Coulter) and MP-count beads (6607115, Biocytex).

## Preparation of murine washed platelets

Wild Type CR57Bl/6J mice (5 to 9 weeks old) were obtained from Janvier-Lab (Le Genest-Saint-Isle, France). Mice were housed as recommended by the European Community Guidelines (directive 2010/63/UE). The animals were housed with enrichment materials, such as Neslets, and provided with food and water freely available. The experimental procedure was approved by the Marseille Ethical Committee #14 (protocol number: APAFIS#20334-2019041811535225). Blood was collected on citrated buffer (1/9e) from anesthetized mice (100 mg/kg of ketamine, 12.5 mg/kg of xylazine and 0.25 mg/kg of atropine). Following blood sampling, while still under deep anesthesia, the animals were euthanized through cervical dislocation. Blood samples were centrifuged at 120 g for 10 minutes at 30°C to obtain platelets rich plasma (PRP). PRP was then centrifuged at 400 g for 10 minutes at 30°C to obtain platelet pellets. Pellets were gently resuspended in Tyrode buffer (133 mM NaCl, 2.7 M KCl, 11.9 mM $NaHCO_3$, 0.36 mM $NaH_2PO_4$, 5 mM glucose, 10 mM Hepes, pH 7.4) supplemented with apyrase (20 U/mL) and PGI2 (1 mM). Then, platelets were washed twice in Tyrode buffer and resuspended in Tyrode buffer with calcium (2 mM) and magnesium (1 mM) and without apyrase and prostacyclin. The washed platelets were kept at 37°C for 30 minutes before use. Platelets were counted by flow cytometry using an antibody directed against GPIb coupled with the dylight 649 (X-649, Emfret) and MP-count beads (6607115, Biocytex).

## Interactions between cancer cells and platelets

Washed human or mouse platelets were added to the cancer cells at a ratio of 50:1 (platelets to cancer cells [11]) in a respective prewarmed FCS-free medium and this interaction was allowed to persist for 3 hours. As a negative control, cancer cells were incubated in FCS-free

non-complete medium. Interaction was stopped by extensive washing and the suitable pre-warmed complete medium was added to the cancer cells. Interaction supernatants were then collected after 3, 21 or 45 additional hours, finally samples were snap frozen and stored at −80°C. During the same time periods, cancer cells behavior after their interaction with platelets were recorded and/or retrieved to analyze proteins composition, glycosyltransferases activities or mRNA contents.

## Holotomography microscopy

Nanolive CX-A microscope was used to analyze the live response of PANC1 or PANCO2 cells during or after their interaction with platelets. For both experiments, $7.10^4$ cancer cells were seeded in 35 mm dishes. Cells were recorded for at least 24 hours at 6 or 12 -minute intervals with or without platelets. All experiments were performed in triplicate. Results were analyzed using EVE Analytics and STEVE software (Nanolive). The cell dry mass is calculated using the *Analysis Standard* module, which allows the dry mass of a cell to be estimated from its refractive index, based on the methods described by Philips *et al.* and Sandoz *et al.* [15,16]. To determine the percentage of cell death, maximum intensity projection images are segmented to isolate individual cells and 320 features - such as shape, texture, and signal distribution - are calculated for each cell. The *Live Cell Death Assay* module, based on a machine learning algorithm a subset of IA, then compares these calculated features with predefined cell death signatures to assess and calculate the probability that a cell is alive or dead.

## Western blot analysis

Cell lysates were obtained with RIPA buffer (50 mM Tris-HCl pH7.4, 150 mM NaCl, 1% Triton x-100, 1% sodium deoxycholate, 0.1% SDS, 1 mM EDTA) supplemented by a cocktail of protease (04693116001, Roche) and phosphatase inhibitors (88667, Thermofisher Scientific). The cell-RIPA mixture was collected and incubated at 4°C for 20 minutes. Protein lysate was obtained after centrifugation at 10,000 rpm for 30 minutes at 4°C and stored at −20°C until used. Protein concentrations were quantified using the Pierce BCA Protein Assay Kit (23225, Thermofisher Scientific). Equal amounts of protein per sample were separated on 4 to 20% Tris-glycine gradient gels (XP04200BOX, Thermofisher Scientific) and transferred to nitrocellulose membranes. Each membrane was incubated in blocking buffer (Tris-HCl pH8 50 mM, NaCl 150 mM, 1% PVP K40, MnCl2 1 mM) for 1 hour and then with primary antibodies or lectins for 2 hours at room temperature.

Primary antibodies used were directed against a human polypeptide N-acetylgalactosaminyltransferase 3 (GALNT3, AF7174, Bio-techne, 0.5 µg/mL), a human fibronectin (FN1, 26836, Cell Signaling Technology, 1/1000) and a mouse FN1 (Ab199056, Abcam, 1/1000).

10 µg/mL of different biotin-linked lectins were used to detect glycosylated structures such as *Vicia villosa* lectin (VVL/VVA, 21510196, Glycomatrix), *Aleuria aurantia* lectin (AAL, MBS656584, MyBioSource), Concanavalin A (ConA, GTX01504, Genetex) and *Phaseolus vulgaris* lectin (PHA-E, 30330017, Glycomatrix).

For both antibodies and lectins, reactivity was determined by fluorescence. This was achieved by using Alexa Fluor-conjugated secondary antibodies (A21102, Thermofisher Scientific, 1 µg/mL and A32733TR, Thermofisher Scientific, 0.2 µg/mL) or Alexa Fluor-conjugated streptavidin (21374, Thermofisher Scientific, 1 µg/mL), respectively. In addition, an anti-GAPDH antibody (2118, Cell Signaling Technology, 1/2000) was used as a control for protein loading and detected with an Alexa-Fluor 647-conjugated secondary antibody (A32733TR, Thermofisher Scientific, 0.2 µg/mL). Images were then acquired and quantified using the iBright FL1500 and iBright Analysis software.

## Immunofluorescence

Briefly, $2.10^5$ PANCO2 cells were seeded in 24-well plates. 24 hours after seeding, cells were fixed in 4% paraformaldehyde for 15 minutes, treated with 0.5% Triton X-100 for 20 minutes at room temperature, followed by blocking with PVPK40 1% for 20 minutes at room temperature. Cells were then immunostained with 2 µg/mL of an antibody directed against GALNT3 (AF7174, Bio-techne) or with the corresponding irrelevant antibody (31243, Thermofisher Scientific) for 2 hours at room temperature. After washing and 45 minutes of incubation with an appropriate secondary antibody (A21102, Thermofisher Scientific), the DNA was stained with Hoechst 33342 (H1399, Thermofisher Scientific) for 10 minutes at room temperature. 10 random, non-overlapping pictures of each well were captured using a Leica DMi8 fluorescence microscope (40X objective).

## Flow cytometry

Analysis of GALNT3 or fibronectin expression on human or mouse pancreatic cancer cells was performed using a flow cytometer (Gallios, Beckman Coulter) with 2 µg/mL of appropriate antibodies. A least 10,000 cells were analyzed for each condition.

## Glycosyltransferase activity test

BxPC-3 and Capan2 cell lysates were assayed for GALNT activity using the Glycosyltransferase Activity Kit according to the manufacturer's instructions (EA001, Bio-techne). GT activity is measured indirectly by employing Malachite Green to detect the release of inorganic phosphate. This release is facilitated by the phosphatase ENTPD3/CD39L3, which quantitatively removes inorganic phosphate from the leaving nucleotide diphosphate, such as UDP or GDP, generated during GT reactions. The Malachite Green signal was then measured using a Thermo Scientific™ Multiskan™ FC microplate photometer at an absorbance wavelength of 620 nm.

## RNA extraction

After 3 hours of incubation/interaction of cancer cells with or without platelets RNA extraction was performed using Trizol (15596026, Thermofisher Scientific). Following 5 minutes on ice, chloroform was added at a volume of 0.2 mL per 1 mL of Trizol used. The solution was then centrifuged, and the aqueous phase containing total RNA was carefully collected. Isopropanol was added at a ratio of 0.5 mL for every 1 mL of Trizol, and the tubes were subsequently placed at −80°C for a minimum of 12 hours. The total RNA was centrifuged to remove the supernatant and was then resuspended in 70% ethanol at a volume of 0.5 mL per 1 mL of Trizol used. To verify protein and solvent contamination of the extracted total RNA, we measured the A260/A280 and A260/A230 ratios using a NanoVue Plus spectrophotometer (Biochrom). Samples with ratios between 1.95 and 2.10 were used. Finally, the total RNA was stored at −80°C until further use.

## Reverse transcription, real time PCR (RT-qPCR)

The pre-incubation of reverse-transcription was performed with 2 µg of total RNA in presence of Oligod (T) 20 primer 50 µM and dNTP 10 mM at 65°C for 5 minutes. Then, Superscript IV Reverse Transcriptase 200 U/uL, DTT 100 mM, RNase out (RNase inhibitor) and SuperScript IV Buffer were added to the total RNA and incubated at 52°C for 10 minutes and at 80°C for another 10 minutes. The resulting cDNA was analyzed using the NanoVue Plus spectrophotometer. qPCR was performed on cDNA samples to compare gene expression between cancer

cells alone and cancer cells after interaction with platelets. The qPCR was performed using a Step One Plus qPCR machine. Samples were tested on preconfigured 96-well qPCR plates (Human glycosylation – 4413255, Human Inflammation - 4418851 or Human tumor metastasis – 4418743, Thermofisher Scientific), with 100 ng added to each well. The microplate was then subjected to the following qPCR program: 50°C for 2 minutes, 95°C for 10 minutes, and then 40 cycles of 95°C for 15 seconds and 60°C for 1 minute. The results were analyzed using Step One software.

### Scratch wound healing assay

PANCO2 cells were plated at $2.10^5$ cells into a 24-well plate and incubated at 37°C 5% $CO2$ until a confluent monolayer was formed (24 hours). After 3 hours of interaction or not with platelets, a 100 μL pipette tip was used to make a wound in the center of the cell monolayer. Cells were then gently washed three times with $PBS^{-/-}$ to remove detached cells. Cells were replenished with complete medium and pictures were captured using an Evos M5000 microscope at T = 0h and T = 24h (10X objective). Pictures were analyzed using FIJI ImageJ software.

### Cell migration test (Boyden chamber)

$8.10^4$ PANCO2 or PANCO2 low FN1 cells were placed in 200 μL of RPMI FCS-free medium on an 8 μm pore diameter Boyden chamber insert in a 24-well plate. The lower chamber was filled with 600 μL of either RPMI medium with or without FCS (negative control). After 16 hours of incubation at 37°C 5% $CO2$, cells in the upper membrane were removed and those attached to the lower membrane were fixed with 2% paraformaldehyde and stained with Hoechst 33342 for 10 minutes at room temperature. The number of migrating cells was assessed by taking 10 random, non-overlapping images of each well using an Evos M5000 microscope (20X objective).

### Multiplex ELISA workflow

After interaction with platelets or not, PANCO2 cell lysates were prepared and supernatants were collected as previously described. Protein concentrations were then quantified using the Pierce BCA Protein Assay Kit (23225, Thermofisher Scientific). Samples were loaded onto Codeplex chips precoated with 16 cytokine capture antibodies. These cytokines are part of the "murine innate immune" or "murine inflammation" panel. The chips were therefore placed in the IsoSpark-DUO machine along with all the necessary reagents according to the manufacturer's instructions (PhenomeX). After 8 hours of running for 4 chips simultaneously, the fluorescence signals were analyzed by IsoSpeak software.

### Statistics

Significance was determined by T-test or ANOVA if the data followed a normal distribution and according to whether 2 or more conditions were to be compared together. If the data did not follow a normal distribution, the Mann-Whitney test was used. Bar graphs are shown as MEAN + SEM.

### Data access for research purposes

Blood samples were anonymized by EFS (Etablissement Français du Sang). We do not have access to patient data.

## Results

### Education by platelets affects the expression of glycosyltransferases in cancer cells

In a previous study, we have demonstrated that platelets were able to transfer their material to cancer cells, highlighting the notion of platelet-educated cancer cell [11]. To determine the cellular consequences of this education on cancer cell behavior, we studied platelet-cancer cell interaction in real-time in "untouched conditions" by holotomographic microscopy coupled with machine learning algorithm, a subset of IA (Fig 1A). To note, the segmentation program was set-up to exclude isolated platelets from the analysis (Fig 1B **and** S1 Fig). Following 48 hours of interaction with platelets, the phenotype of cancer cells was affected and become more proliferative (Fig 1C**; upper panel**). Interestingly, the presence of platelets significantly increased the total dry mass of proliferative cancer cells (Fig 1C**; middle panel**), as well as the dry mass in each newly formed cell (Fig 1C**; lower panel**). These results suggest that a compositional change occur in cancer cells overtime, either by direct transfer of material from platelets to cancer cells or by indirect stimulation of intrinsic signaling pathways leading to an increase of intracellular compounds.

To determine if this change in dry mass was due to a transfer of material, i.e., of proteins, from platelets to cancer cells, we next focused on the possible transfer of glycosyltransferases (GT) between platelets and cancer cells. Platelets were indeed previously described to express and to secrete GT. We focused our analysis on the GALNT family, which catalyzes the first O-glycosylation reaction and includes 20 different members. We first determined the GALNT activity in platelets and cancer cells prior interaction (Fig 1D). Unexpectedly, a very low specific rate of catalysis of GALNT family was observed in washed platelets compared to cancer cells. These results were confirmed by focusing on GALNT3 main member of the GT family. We confirmed the absence of GALNT3 in platelets, using western Blot (Fig 1E) and flow cytometry (Fig 1F) in comparison with cancer cells (Fig 1G). Taken together, these results indicate that platelets do not express functional GALNT proteins.

We next studied the variation in the expression of mRNAs encoding for 43 different GT families in platelet-educated cancer cells compared to cancer cells alone. We observed strong differences in the mRNA expression levels encoding for the different GT families studied in platelet-educated cancer cells compared to cancer cells alone (Fig 2A). Of interest, 3 GT families stood out in cancer cells after their interaction with platelets: MGAT2, MGAT3 and B3GNT which were significantly overexpressed by 343, 320 and 170%, respectively (Table 1). These results appear to be confirmed at the protein level, with a trend towards increased expression of MGAT2 and of MGAT2-catalysed motifs recognized by the lectin PHA-L only 3 hours after interaction of cancer cells with platelets (S2 **and** S3 Figs). In line with these mRNA variations, we also observed a change in glycosylation patterns using different lectins (Figs 2B). The Tn antigen expression (recognized by VVL/VVA) at the surface of cancer cells is significantly decreasing in platelet-educated cancer cells (Fig 2B**; left upper panel**) in comparison with cancer cells alone, while fucosylation (revealed by AAL) is significantly increased (Fig 2B**; right upper panel)**. Nevertheless, there is no change in the expression of α-mannosyl or α-glucosyl residues recognized by ConA (Fig 2B**; left lower panel)** or bisecting GlcNAc and galactose residues detected by PHA-E (Fig 2B**, right lower panel**), in platelet-educated cancer cells. Altogether these results indicate that the education by platelets affects the expression of several GT in cancer cells. Among them, the overexpression of B3GNT may contribute to the hyperglycosylation observed in cancer cells. Knowing that this GT family are implicated in the formation of carbohydrates involved in metastasis, such as poly-N-acetyllactosamine, their overexpression may affect cancer cell behavior.

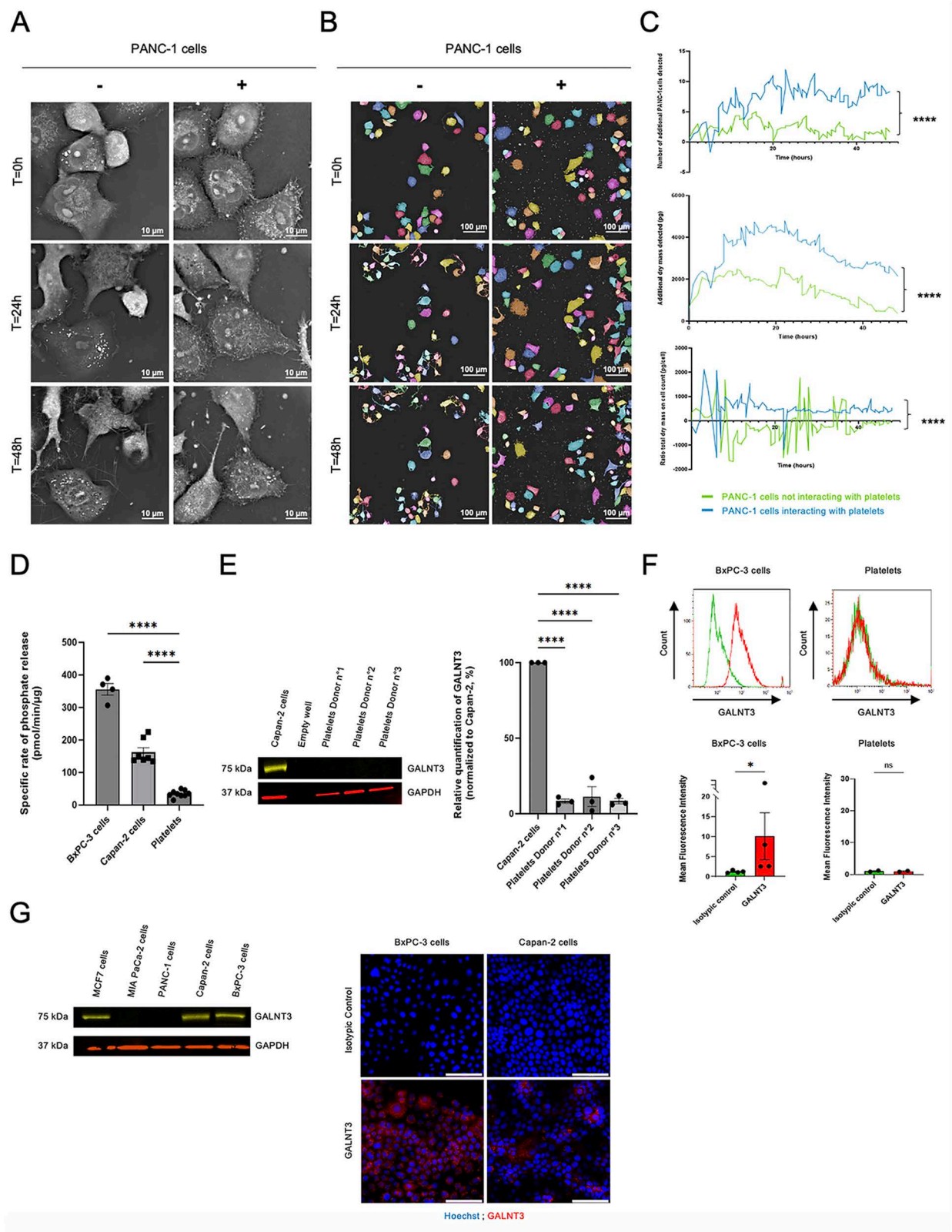

**Fig 1. Platelets increase material content in human pancreatic cancer cells during their interaction and express low levels of the GALNT family. (A)** Representative images of three independent experiments of PANC-1 cells, in the presence (+) or in absence (−) of platelets over time

up to 48 hours (holotomographic microscopy bars, 10 μm). **(B)** Representative images of three independent experiments of PANC-1 cells in the presence (+) or absence (−) of platelets over time up to 48 hours (holotomographic microscopy bars, 100 μm). Segmentation was performed using the "Standard Analysis" module, excluding platelets from the analysis. **(C)** Graph was obtained with "Standard Analysis" module showing the increase of newly synthesized cells **(upper panel)**, the accumulation of dry mass **(middle panel)** or the ratio of total dry mass per cell **(bottom panel)** over time in presence (blue) or in absence (green) of platelets. Each curve represents the mean of three independent experiments (Mann-Whitney test, ****$P < 0.0001$). **(D)** Evaluation of GALNT family activity in pancreatic cancer cells (BxPC-3 and Capan-2) and in human platelets (three independent donors) using an indirect colorimetric glycosyltransferase activity assay. Graph depicts the mean value (+/− SEM) from three independent experiments of the rate of phosphate release corresponding to the specific activity of the GALNT family (including 20 members). One-way ANOVA, ****$P < 0.0001$. **(E)** Representative image **(left panel)** of GALNT3 detection by Western Blot in 50 μg of lysate of Capan-2 cells or human platelets (three independent donors). Detection of GAPDH was used as a loading control for the experiment. Histogram **(right panel)** represents the relative quantification of the GALNT3/GAPDH ratio in human platelets normalized to Capan-2 cells (mean +/− SEM) of three independent experiments (ANOVA test, ****$P < 0.001$). **(F)** Representative graphs of indirect GALNT3 detection **(upper panels)** using unconjugated anti-human GALNT3 antibody (2 μg/mL) and AF680-conjugated anti-sheep IgG secondary antibody (2 μg/mL) by flow cytometry in BxPC-3 cells (red curve, left upper panel) or human platelets (red curve, right upper panel) compared to isotype control (2 μg/mL) revealed with AF680-conjugated anti-sheep IgG secondary antibody in both cases (green curves, all panels). Histograms show **(lower panels)** mean of fluorescence intensity (+/− SEM) of GALNT3 expression in BxPC-3 cells and in human platelets assessed from two or four independent experiments (Mann-Whitney test, *$P < 0.0286$, ns = not significant). **(G)** Representative image of GALNT3 (75 kDa) detection by Western Blot **(left panel)** in 50 μg of lysate of four different human pancreatic cancer cell lines. MCF7 cell line was used as positive control. Detection of GAPDH was used as a loading control for the experiment. Representative images **(right panel)** of indirect GALNT3 (red) detection by immunofluorescence in BxPC-3 and Capan-2 cells using unconjugated anti-human GALNT3 antibody (2 μg/mL) and AF680-conjugated anti-sheep IgG as a secondary antibody (2 μg/mL) compared to isotype control (2 μg/mL) revealed with AF680-conjugated anti-sheep IgG secondary antibody. Cell nuclei (blue) were stained with Hoechst. Images were taken with a 40X objective, 5 images per sample, bars 100 μm. These cell lines were then used as positive controls for GALNT3 expression.

## Cancer cells educated by platelets overexpress 40 genes involved in metastasis and inflammation in cancer cells

We then studied if the education by platelets may affect the expression of various RNAs other than GT. We therefore analyzed the presence of mRNAs for 124 other genes encoding proteins involved in metastasis and inflammation. We observed by RT-qPCR that about 40 genes were overexpressed by more than 100% in platelet-educated cancer cells compared to cancer cells alone (Fig 2C). Among these genes, the most overexpressed in platelet-educated cancer cells are those encoding proteins involved in cancer progression and metastasis, including SERPINE1 (358%), MMP2 (297%), MMP10 (214%), TIMP1 (187%), FN1 (166%), TMPRSS4 (146%) and RHOC (102%) (Table 1). However, we also observed that there are mRNAs coding for CTBP1 (184%), APC (107%), and TP53 (104%) proteins, which are all tumor suppressors, that repair DNA damage, regulate cell cycle and apoptosis and so prevent cancer initiation and progression (Table 1). Taken together, these two mRNA clusters highlight the dual role of platelets in tumor progression. Finally, we have identified mRNAs encoding proteins such as IL2RG (155%), GNRH1 (153%) and CACNA1D (130%) that are overexpressed and whose role in cancer remains unclear (Table 1). For example, mRNA encoding IL2RG, the gamma subunit common to the IL-2, IL-4, IL-7 and IL-21 receptors, is overexpressed by 155% in platelet-educated cancer cells (Table 1). All these receptors are crucial for various functions, including immune development and regulation, inflammation, and cell proliferation.

To get a more precise view of the protein-protein interaction network of these genes we next used the STRING database. We classified only the 38 genes with an 100% overexpression in platelet-educated cancer cells into 5 gene clusters according to their biological function: (1 - red) ECM degradation, adhesion, migration and invasion (2 – yellow) tumor suppressors (3 – green) proliferation, motility and regulation of cell death (4 – dark blue) anti-metastatic and (5 – light blue) genes with no common function (Fig 2D). Within the main cluster in red, one glycoprotein appeared to be central, fibronectin (FN1) which is overexpressed by 166% in platelet-educated cancer cells (Fig 2C). This glycoprotein is known to play a key role in cell adhesion and migration, and thus in the development of metastasis (Table 1). We confirmed

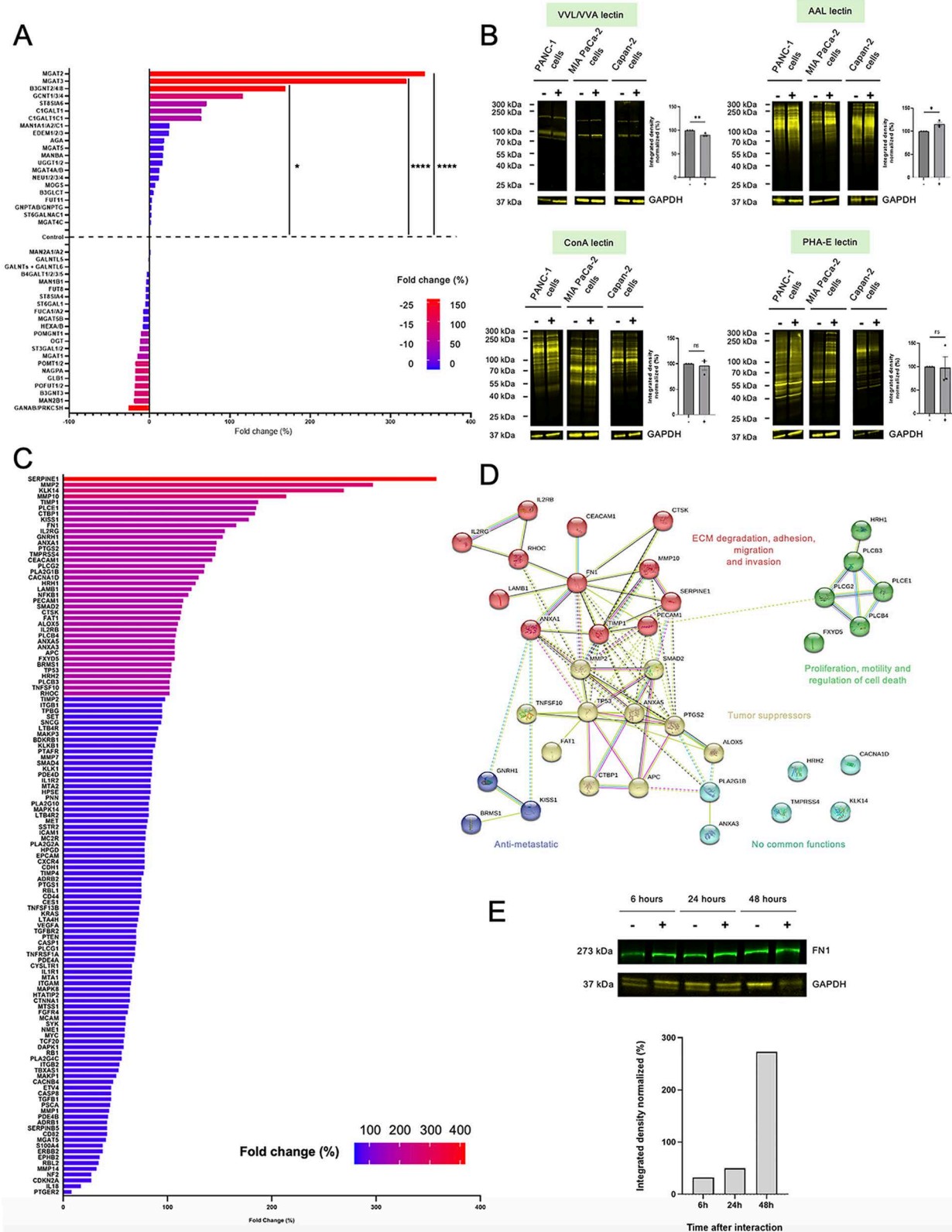

**Fig 2. Effects of Platelet-Cancer Cell Interaction on the Expression of Glycosyltransferases, Inflammation, and Metastasis-Related Genes.**
(**A**) Graph shows the mean fold change (percentage) of mRNA expression of 43 GT families in pancreatic cancer cells (PANC-1, MIA PaCa-2,

Capan-2) after 3 hours of interaction of cancer cells with platelets compared to before their interaction (control). Colors represent an overexpression of at least 150% (red), 100% (pink), 50% (purple) or less than 50% (blue) of the mRNA of GT families in cancer cells after platelet interaction or, conversely, an underexpression of at least 25% (red), 15% (pink), 10% (purple) or less than 10% (blue). Data were normalized to the housekeeping gene HPRT1 and histogram represents the mean of three independent experiments (ANOVA test, ****$P < 0.0001$, *$P = 0.0401$). **(B)** Representative images and corresponding quantification of glycan motif detection using 4 lectins: VVL/VVA, **(left upper panel)** AAL **(right upper panel),** ConA **(left lower panel),** PHA-E **(right lower panel)** by Western Blot of pancreatic cancer cell lysates (PANC-1, MIA PaCa-2 or Capan-2, 40 μg) interacting (+) or not (−) with platelets for 3 hours. GAPDH expression was detected as a loading control for the experiment. The graphs show the mean of integrated density (+/− SEM) of glycan motif expression normalized to cancer cells alone (t-test, **$P = 0.0086$, *$P = 0.0240$, ns = not significant). **(C)** The graph represents the mean fold change (percentage) of mRNA expression of 124 genes encoding proteins involved in inflammation and metastasis in a human colorectal cancer cell line (HT-29) after 3 hours of interaction with platelets compared to before their interaction. Colors represent an overexpression of at least 300% (red), 200% (pink), 100% (purple) or less than 100% (blue) of the mRNA in platelet-educated cancer cells. Data were normalized to the housekeeping gene HPRT1. **(D)** Protein-protein interaction network using STRING database 12.0. Only the 38 genes with at least an 100% overexpression were considered to perform this analysis. Nodes represent proteins and each node color (red, yellow, dark blue, light blue or green) indicates a gene cluster. Edges represent known interactions from the curated database (light blue) or experimentally determined (pink); scientific literature (yellow/green); co-expression (black) or protein homology (dark blue). **(E)** Representative image of fibronectin (FN1, 273 kDa, left panel) detection by Western Blot in 15 μg of PANCO2 cell lysates 6, 24 or 48 hours after their interaction with platelets (+) or not (−). GAPDH expression was detected as a loading control. The histogram (down panel) presents the relative quantification of the FN1/GAPDH ratio in PANCO2 cells + platelets, normalized to PANCO2 cells alone, over time up to 48 hours.

this result by Western-blotting, demonstrating an overexpression of FN1 in platelet-educated cancer cells (Fig 2E). This overexpression was observed as early as 6 hours following the interaction with platelets (Fig 2E). Since FN1 is highly glycosylated and involved in many aspects of cancer metastasis, we next focused on it.

## The case study of fibronectin: Its overexpression affects cancer cell behavior

We used miRNA transfection to generate a murine pancreatic cancer cell line that underexpresses FN1 by 2-fold (Fig 3A-B), referred to as PANCO2 low FN1. Our first step was to examine, using untouched holotomographic microscopy (Fig 3D), the effect of FN1 underexpression on cancer cell death. No difference between PANCO2 and PANCO2 low FN1 cell was detected, suggesting that FN1 doesn't play a role in this process (Fig 3E). Cancer cell migration was evaluated by wound healing assay (Fig 3F). Results obtained indicate that PANCO2 low FN1 cells migrate faster than PANCO2 cells. This result was confirmed using a Boyden chamber (Fig 3G). These data indicate that the level of expression of FN1 in cancer cells affects the migration of cancer cells.

Next, we wanted to access the effect of platelets on cancer cells. To do this, we allowed cancer cells to interact with platelets for 3 hours. After extensive washes, we let cancer cell to grow for 3, 24 or 45 additional hours (Fig 4A). Using holotomographic microscopy, we evaluated the percentage of cell death (Fig 4B-C). As a result, we found no differences between cancer cells that interacted with platelets and those that did not. Thus, platelets do not seem to be involved in cancer cell death. We also evaluated cancer cell migration by wound healing assay (Fig 4D). Interestingly, we observed that PANCO2 cells that didn't interact with platelets and PANCO2 low FN1 cells that interact with platelets had a similar migration. This means that the previously observed migratory difference between these two cell types was compensated in the presence of platelets. These data highly suggested that platelets can stimulate FN1 production in cancer cells. This result was confirmed by Western blot analysis. We observed an 80% increase in FN1 expression in PANCO2 low FN1 cells 48 hours after their interaction with platelets (Fig 4E).

To determine how the education of platelets could lead to the overexpression of FN1, we next assessed the expression of cytokines in PANCO2 low FN1 cells 6 and 24h (Fig 4F) after their interaction with platelets using PhenomeX technology. Among

**Table 1. Summary of genes over-expressed by at least 100% in platelet-educated cancer cells compared to cancer cells alone.**

| Genes | Complete Name | % of overexpression in platelet-educated cancer cells vs. cancer cells alone | Functions | Type of cancer in which its involvement has been demonstrated |
|---|---|---|---|---|
| SERPINE1 / A1PI / A1AT | Serpin Family E Member 1 / alpha-1 proteinase inhibitor / alpha-1 antitrypsin | 358 | Serine protease inhibitor, Protecting lung damage tissue, Anti-inflammatory, Immune responses regulation, Cancer metastasis | Breast, lung, colorectal, ovarian, gastric, kidney, bladder, melanoma and glioblastoma |
| MGAT2 / GnT-II | Mannosyl (Alpha-1,6-)-Glycoprotein Beta-1,2-N-Acetylglucosaminyltransferas | 343 | Catalyses the addition of N-acetylglucosamine to mannose, part of the common structure of N-glycan complexes and hydrides - Stability, function and localization of proteins | Breast, ovarian, neuroblastoma, pancreas (current study) |
| MGAT3 / GnT-III | Mannosyl (Beta-1,4-)-Glycoprotein Beta-1,4-N-Acetylglucosaminyltransferase | 320 | Adds an N-acetylglucosamine to the central mannose of the common structure of complex and hydride N-glycans - Stability, function and good localization of proteins | Breast, colorectal, melanoma, pancreas (current study) |
| MMP2 | Matrix MetalloProteinase-2 / Gelatinase A | 297 | Extracellular matrix degradation, Tissue remodeling, Tissue repair and regeneration, Angiogenesis, Inflammation, Cardiovascular health, Neurological function, Cancer metastasis | Breast, lung, colorectal, ovarian, prostate, glioma, melanoma and pancreatic |
| KLK14 | Kallikrein 14 | 269 | Proteolytic activity (serine protease), Tissue remodeling, Inflammation, Skin barrier function, Salivary gland function, Cancer | Breast, lung, colorectal, ovarian, prostate and pancreatic |
| MMP10 | Matrix MetalloProteinase-10 / Stromelysin-2 | 214 | Extracellular matrix degradation, Tissue remodeling, Tissue repair and regeneration, Angiogenesis, Inflammation, Cardiovascular health, Neurological processes, Cancer metastasis | Breast, lung, colorectal, ovarian, pancreatic, gastric and oral |
| TIMP1 | Tissue Inhibitor Metallo-Proteinases 1 | 187 | MMP inhibitor, Tissue homeostasis, Wound healing, Angiogenesis, Anti-inflammatory, Neurological function, Cancer metastasis | Breast, lung, colorectal, ovarian, prostate, pancreatic, gastric, bladder, head and neck |
| PLCE1 | Phospholipase C Epsilon 1 | 185 | Phospholipase activity, Intracellular Signaling, (calcium signaling, cell proliferation, cardiovascular and immune functions), Development, Cancer | Lung, colorectal, gastric and esophageal |
| CTBP1 | C-Terminal Binding Protein 1 | 184 | Transcriptional corepressor, Epigenetic regulation, Cellular signaling, Cellule differentiation, proliferation (cell cycle regulation) and apoptosis, Metabolic regulation, DNA damage response, Development, Tumor suppression | Breast, lung, colorectal, bladder and liver |
| KISS1 | Kisspeptin-1 | 178 | Puberty initiation, Gonadotropin regulation, Reproductive cycle regulation, Pregnancy and placental function, Metabolism, Cancer | Breast, colorectal, ovarian, prostate, gastric, cervical and endometrial |
| B3GNT2/4/8 | UDP-GlcNAc: BetaGal Beta-1,3-N-Acetylglucosaminyltransferase | 169 | On Core 2 N-glycans and O-glycans, they transfer an N-acetylglucosamine onto a Galactose which creates a structure called "poly-N-acetyllactosamine" which has 3 main functions:<br>- Provide a base structure for further modifications<br>- Serve as a ligand for antibodies and lectins, modulating the function of the proteins that carry it, and even the fate of cells.<br>- Sometimes masks the original structures of sugars and helps cancer cells evade the immune system. | Breast, lung, colorectal, ovarian, pancreas (current study) |
| FN1 | **Fibronectin 1** | **166** | **Cell adhesion, Wound healing, Tissue development, Cell migration, Angiogenesis, Inflammation, Cell signaling (proliferation, survival, gene expression), Stem cell differentiation, Support of tissue and organs, Cancer metastasis** | **Breast, lung, colorectal, ovarian, prostate, melanoma, pancreatic, gastric and liver** |
| IL2RG | Interleukin-2 Receptor Subunit Gamma | 155 | Interleukin-2 signaling (T cell and NK cell development), Immune regulation, Cytokines signaling (immune regulation, proliferation, differentiation) | – |
| GNRH1 / LHRH | GoNadotropin-Releasing Hormone 1 / Luteinizing Hormone-Releasing Hormone / Gonadorelin | 153 | Gonadotropin release and sex hormone production, Puberty Onset, Menstrual cycle regulation | – |

*(Continued)*

**Table 1.** (Continued)

| Genes | Complete Name | % of overexpression in platelet-educated cancer cells vs. cancer cells alone | Functions | Type of cancer in which its involvement has been demonstrated |
|---|---|---|---|---|
| ANXA1 | Annexin A1 / Lipocortin-1 | 147 | Phospholipids binding, Anti-inflammatory, Cell membrane function, Tissue repair and wound healing, Immune system regulation Cardiovascular health, Neurological function, Cancer | Breast, lung, colorectal, ovarian, prostate, pancreatic, gastric, bladder, head and neck |
| PTGS2 / COX-2 | Prostaglandin-Endoperoxide Synthase 2 / CycloOygenase-2 | 147 | Inflammation (prostaglandins production, pain sensation, fever), Gastric protection, Reproductive health, Tissue repair, Cardiovascular health, Neurological function, Cancer | Breast, lung, colorectal, ovarian, prostate, brain, pancreatic, gastric, bladder, liver, head and neck, esophageal and skin |
| TMPRSS4 | TransMembrane Protease Serine 4 | 146 | Proteolytic activity (serine protease), Inflammation, Cell signaling, Cancer metastasis | Breast, lung, colorectal, ovarian, pancreatic, gastric and liver |
| CEACAM1 | CarcinoEmbryonic Antigen-related Cell Adhesion Molecule 1 | 143 | Cell-Cell adhesion, Immune cell regulation, Angiogenesis, Infection, Reproduction, Tissue development, Cell signaling, Cancer | Breast, lung, colorectal, ovarian, pancreatic, prostate and liver |
| PLCG2 | PhosphoLipase C Gamma 2 | 136 | Signal transduction (calcium release and protein kinase C activation), Immune responses, Hematopoiesis, Cell growth and proliferation, Inflammation, Neurological function, Cancer | Leukemia (CLL, CMML) and lymphoma (DLBCL, PTCL, CTCL) |
| PLA2G1B | PhosphoLipase A2, Group IB | 135 | Digestive enzyme (lipid metabolism), Cell signaling (arachidonic acid), Inflammation, Immune responses, Phospholipids turnover | Breast, colorectal, ovarian and pancreatic |
| CACNA1D | Calcium Chanel, Voltage-Dependent, L Type, Alpha 1D Subunit | 130 | Calcium channel formation (voltage sensing), Cardiovascular function, Neuronal signaling, Smooth muscle contraction, Hormone secretion | – |
| HRH1 | Histamine H1 Receptor | 127 | Allergic responses (itch sensation, ocular effects), Inflammation, Angiogenesis, Cell proliferation and growth, Smooth muscle contraction, Central nervous system effects, Appetite regulation, Cardiovascular function | Breast, lung, colorectal, ovarian, thyroid and gastric |
| LAMB1 | Laminin subunit Beta-1 | 123 | Extracellular matrix formation, Cell adhesion, Basement membrane, Tissue organization, Neurological function, Skin integrity, Renal function, Muscle function, Cancer | Breast, lung, colorectal, ovarian, pancreatic and gastric |
| NFKB1 | Nuclear Factor Kappa B Subunit 1 | 120 | Transcription factor, Immune responses, Bacterial and viral infections, Inflammation, Cell survival, Anti-oxidative response, Cell development and differentiation, Metabolism regulation, Neurological function, Cancer | Breast, lung, colorectal, ovarian, prostate, pancreatic, liver, multiple myeloma, lymphomas and leukemias |
| GCNT1/3/4 / C2GnT | Beta-1,3-Galactosyl-O-Glycosyl-Glycoprotein Beta-1,6-N-Acetylglucosaminyltransferase | 116 | In O-glycosylation, GCNTs convert Core 1 into Core 2, giving rise to structures such as SLe$^x$. They also initiate the synthesis of a structure known as "poly-N-acetyllactosamine" | Breast, colorectal, ovarian, pancreas (current study) |
| PECAM1 / CD31 | Platelet Endothelial Cell Adhesion Molecule-1 | 115 | Cell adhesion, Leukocyte transendothelial migration, Platelet function, Thrombosis and hemostasis, Inflammation, Angiogenesis Immune cell trafficking (T cells), Autoimmunity, Cancer | Breast, lung, colorectal, prostate and pancreatic |
| SMAD2 | Mothers Against Decapentaplegic Homolog 2 | 114 | TGF-β signaling (cell growth, proliferation, differentiation, immune responses, wound healing, tissue repair, apoptosis), Embryonic development, Cancer | Breast, lung, colorectal, pancreatic and liver |
| CTSK | Cathepsin K | 113 | Bone resorption (bone remodeling, growth, repair), Extracellular degradation, Connective tissue turnover, Lysosomal function, Periodontal disease and arthritis, Cancer | Breast, lung and prostate |
| FAT1 | Fat Mass and Obesity-Associated Protein | 112 | Cell-cell adhesion, Tissue development and organization (planar cell polarity regulation), Brain and kidney development, Immune responses, Cell signaling, Cardiovascular function, Cancer | Breast, colorectal, glioblastoma, head and neck squamous cell carcinoma |

*(Continued)*

**Table 1.** (Continued)

| Genes | Complete Name | % of overexpression in platelet-educated cancer cells vs. cancer cells alone | Functions | Type of cancer in which its involvement has been demonstrated |
|---|---|---|---|---|
| ALOX5 | Arachidonate-5-Lipoxygenase | 110 | Leukotriene biosynthesis, Inflammation, Asthma and allergic reactions, Cell growth and differentiation, Cardiovascular and neurological functions, Cancer | Breast, lung, colorectal and prostate |
| IL2RB | Interleukin-2 Receptor subunit Beta | 109 | Interleukin-2 signaling (immune system regulation, viral infections, immune tolerance, T cell activation, transplantation), Cancer | Melanoma, kidney, leukemia and T-cell lymphomas |
| PLCB4 | PhosphoLipase C Beta 4 | 108 | Signal transduction (calcium release and protein kinase C activation), Cell proliferation and migration, Immune responses, Cardiovascular and neurological functions, Cancer | Breast, lung, colorectal and ovarian |
| ANXA5 | Annexin A5 / Annexin V | 107 | Phospholipids binding, Apoptosis regulation, Blood coagulation, Anti-inflammatory, Membrane repair, Cell growth and differentiation, Cardiovascular and neurological functions, Cancer | Breast, lung, colorectal, ovarian and prostate |
| ANXA3 | Annexin A3 | 107 | Phospholipids binding, Anti-inflammatory, Thrombosis, Anti-coagulant, Cell homeostasis (cell membrane and cytoskeletal organization, intracellular signaling), Cancer | Breast, lung, pancreatic, gastric and liver |
| APC | Adenomatous Polyposis Coli | 107 | Tumor suppressor gene (Wnt signaling), Cell adhesion and migration, Cancer | Colorectal |
| FXYD5 | FXYD domain-containing ion transport regulator 5 | 107 | Sodium-potassium pumps regulation (renal function), Cancer | Breast, lung and ovarian |
| BRMS1 | Breast cancer Metastasis Suppressor 1 | 104 | Metastasis suppressor | Breast and melanoma |
| TP53 | Tumor Protein P53 | 104 | Tumor suppressor, Cell cycle regulation, DNA repair (genome stability), Apoptosis and senescence, Anti-angiogenic, Metabolism regulation, Immune system modulation, Cancer | Breast, lung, colorectal, ovarian, pancreatic, liver, bladder, head and neck, esophageal and leukemias |
| HRH2 | Histamine Receptor H2 | 103 | Histamine signaling, Gastric acid regulation (digestive role) | Breast, colorectal, pancreatic and gastric |
| PLCB3 | PhosphoLipase C Beta 3 | 102 | Signal transduction (calcium release and protein kinase C activation), Cell proliferation and differentiation, Neuronal signaling, Immune responses, Cancer | Breast, lung, colorectal and ovarian |
| TNFSF10 | Tumor Necrosis Factor ligand SuperFamily Member 10 | 102 | TNF signaling – TRAIL (apoptosis induction, immune responses, inflammation), Cancer | Breast, lung, colorectal, pancreatic and prostate |
| RHOC | Rho GTPase C | 102 | Cell adhesion and migration, Angiogenesis, Cytoskeletal dynamics, Gene expression regulation, Cancer metastasis | Breast, lung, colorectal and melanoma |

21 cytokines tested, the expression of 5 cytokines TNF-α, EGF, PDGF, IL-6 and IFNγ increased during the first 24 hours. These particular cytokines are all known to be involved in FN1 production though different signaling pathways such as MAPK/PI3K/Akt, STAT3 or JAK/STAT1 pathways [17–21]. It should be noted that TNF-α, IL-6 and IFNγ are also known to be pro-apoptotic [22–24]. However, other anti-apoptotic molecules such as GM-CSF, IL-2, and IL-15 are also over-expressed and may counterbalance this effect, leading to no significant difference in cell death between the two lines tested over the 24-hour period.

Taken together our results indicate that platelet-educated cancer cells produce different cytokines involved in the overexpression of fibronectin at their surface, facilitating, in turn, their adhesion.

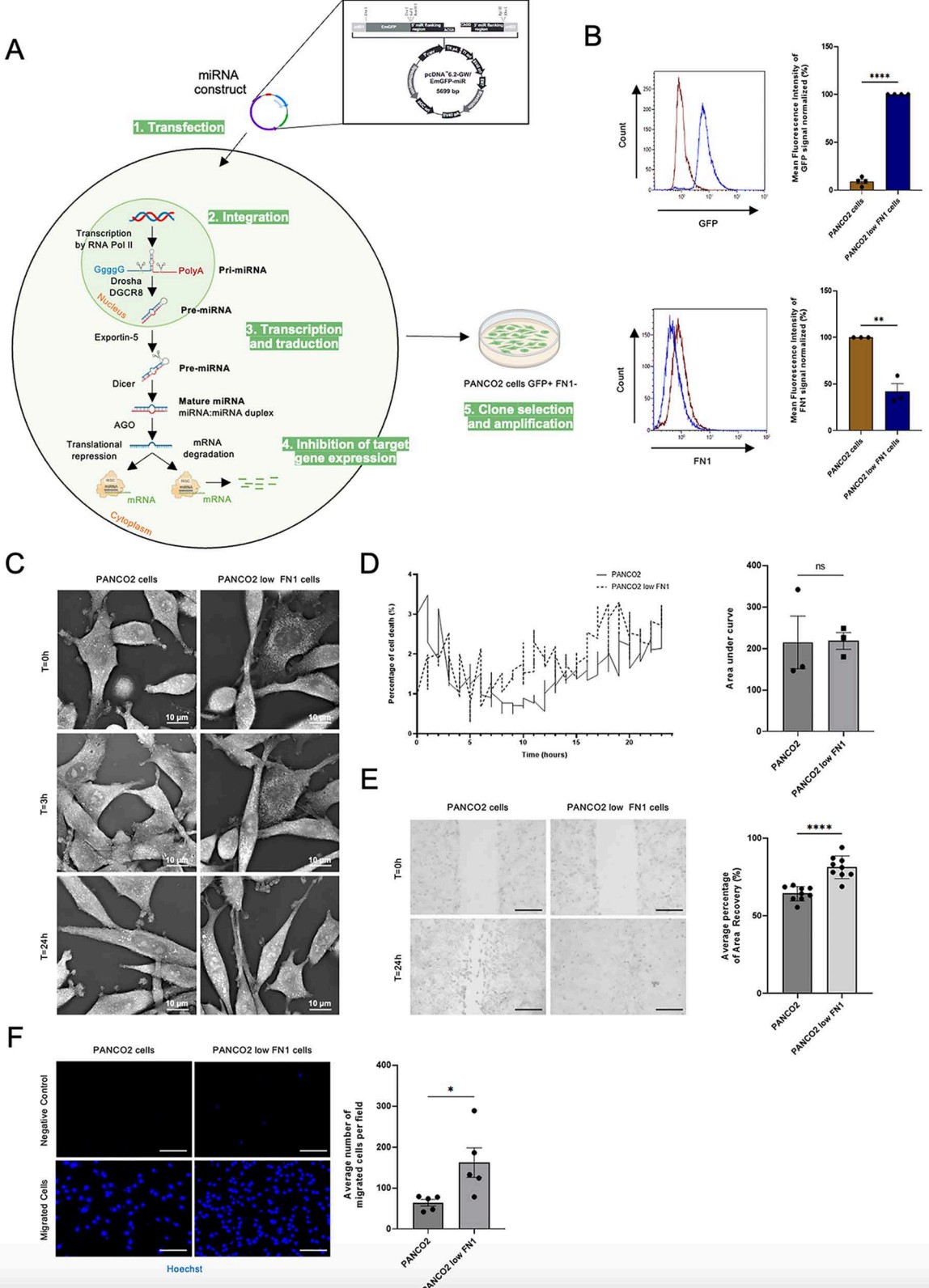

**Fig 3. Overexpression of fibronectin (FN1) in platelet-educated cancer cells reduces cell migration without affecting the death rate.** **(A)** Using miRNA transfection, we generated a murine pancreatic cancer line that underexpress fibronectin, called PANCO2 low

FN1 cells. **(B)** Representative graphs and corresponding quantification of GFP detection **(upper panels)** by flow cytometry in PANCO2 mock (brown) *vs.* PANCO2 low FN1 (blue) cells. The histogram represents the mean value of four independent experiments (t-test, ****P < 0.0001). Representative graphs and respective quantification of FN1 detection **(lower panels)** with unconjugated anti-mouse FN1 antibody (2 μg/mL) and AF647-conjugated anti-rabbit IgG secondary antibody (2 μg/mL) by flow cytometry in PANCO2 mock (brown) *vs.* PANCO2 low FN1 (blue) cells compared to isotype control (2 μg/mL) detected with AF47-conjugated anti-rabbit IgG secondary antibody (brown curves). Histogram shows mean fluorescence intensity (+/− SEM) from three independent experiments (t-test, **P = 0.0025). **(C)** Representative images of three independent experiments of PANCO2 mock or PANCO2 low FN1 cells over time up to 24 hours (holotomographic microscopy bars, 10 μm). **(D)** Curves **(left panel)** were generated with the "Cell Death Assay" module showing the percentage of cell death in PANCO2 mock (solid line) or PANCO2 low FN1 (dashed line) cells over 24 hours. The histogram **(right panel)** shows the mean area under the preceding curve (+/- SEM) of three independent experiments (t-test, ns = not significant). **(E)** Representative images **(left panel)** of nine independent experiments of wound healing assay experiments evaluating PANCO2 mock or PANCO2 low FN1 cell migration over 24 hours (10X objective, bars 125 μm). Graph **(right panel)** shows the mean percentage of area recovery (+/− SEM), from these nine independent experiments, corresponding to the migration of PANCO2 mock or PANCO2 low FN1 cells (5 images per experiment and per well, t-test, ****P < 0,0001). **(F)** Representative images **(left panel)** of five independent migration assays to detect PANCO2 mock or PANCO2 low FN1 migrated cells, with Hoechst 33342 after 16 hours. The negative control was performed with the same cells but without FCS in the lower chamber (20X objective, bars 125 μm). Graph **(right panel)** represents the mean (+/− SEM) of the number of migrated PANCO2 mock or PANCO2 low FN1 cells, from five independent experiments, after 16 hours of incubation (10 images per experiment and per well, t-test, *P = 0.03, n = 5).

## Discussion

In the blood circulation, platelets play an important role in cancer, but their role in the tumor microenvironment, in which they can directly interact with the tumor, is poorly understood. It was first reported in 2012 that extravasated platelets were present in the tumor microenvironment in an ovarian cancer and then in breast cancer and melanoma [10,25]. We have previously shown that 20% of the total platelets present in the tumor microenvironment are found in the periphery and not in the center of the tumor [11]. Their presence is not coincidental and usually correlates with advanced stages of the disease [11]. Several hypotheses have been proposed to explain how platelets can reach the extravascular space and interact directly with cancer cells: (I) following intratumoral bleeding, (II) *via* transmigration directly or indirectly through association with leukocytes, or (III) due to extramedullary hematopoiesis [26]. Until now, it has not been demonstrated how platelets access the primary tumor site. However, several studies have investigated their role within the primary tumor and highlighted their ability to drive phenotypic changes in cancer cells. This may or may not be beneficial for cancer development.

Previously, we showed that platelets can influence cancer cell behavior by educating them, i.e., by transferring material to them (lipids, proteins or RNA) [11,12]. We had demonstrated that platelets were capable of releasing the contents of their α-granules into tumor cells, or generating MPs enabling the transfer of adhesive proteins from one cell type to another [11]. In the present study, we wanted to characterize exactly which genes and proteins are involved in this education of cancer cells by platelets.

To this end, we conducted an *in vitro* study using cell lines derived from human digestive cancers such as colorectal adenocarcinoma (HT29), pancreatic adenocarcinoma (BxPC-3, Capan-2), and pancreatic carcinoma (PANC-1, MIA PaCa-2). Colorectal cancer, associated with a moderate risk of thrombosis, and pancreatic cancer, linked to a higher thrombotic risk, were selected to explore the interaction between cancer cells and platelets. We chose to focus on pancreatic cancer because of its higher mortality rate. To study this interaction in detail, we used four pancreatic cancer cell lines: BxPC-3, Capan-2, PANC-1 and MIA PaCa-2. These cell lines were chosen for their diversity in terms of gender, age, and genetic mutations [27]. All except BxPC-3 (which lacks the Kras mutation) carry the four most common mutations in pancreatic cancer: Kras, TP53, CDKN2A/p16 and SMAD4-DPC4 [27,28]. Notably, HT29 also carries a TP53 mutation, which is common in colorectal cancers [28]. In addition, these cell lines

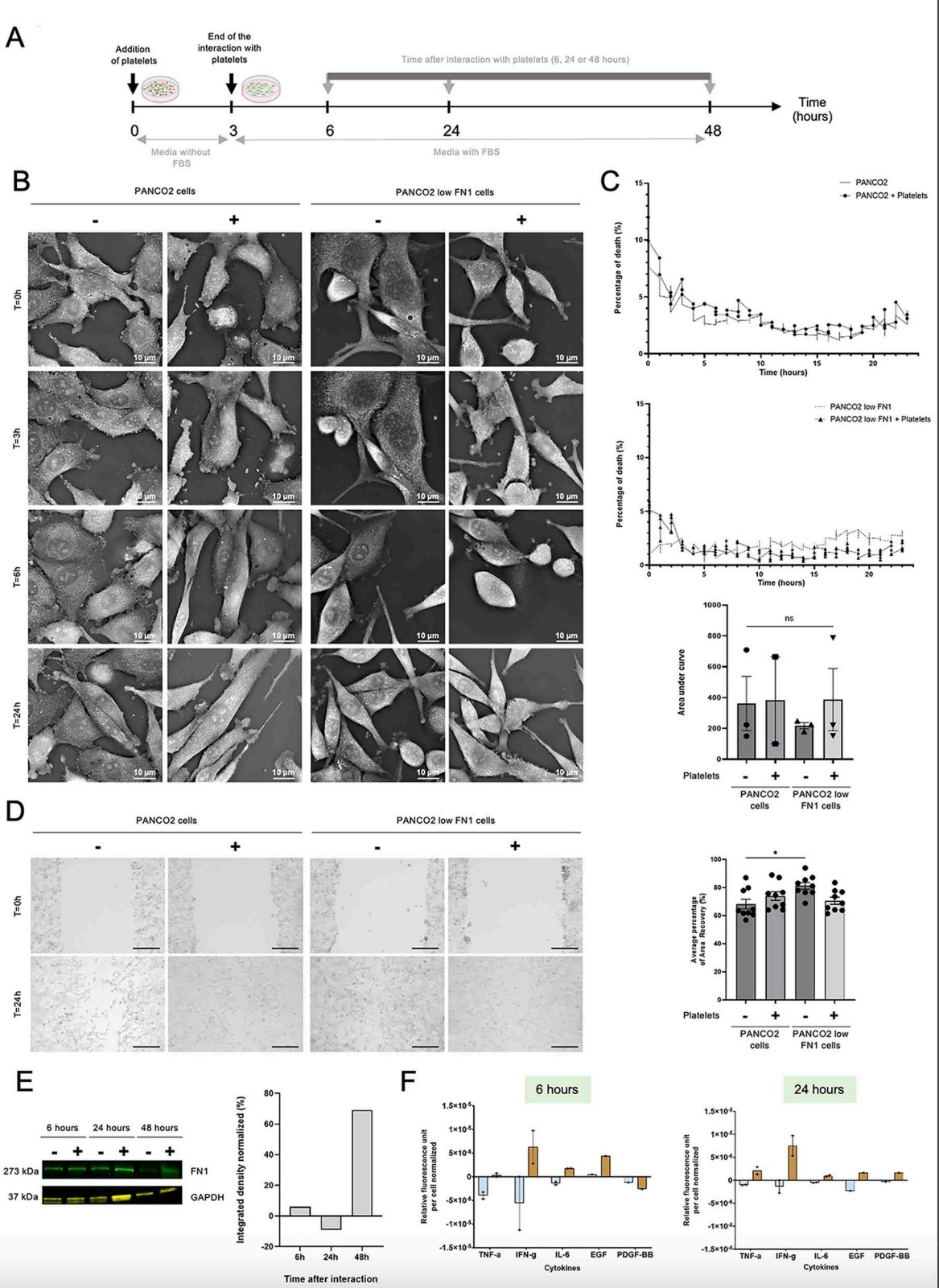

**Fig 4. Platelets stimulate FN1 production in cancer cells. (A)** Schematic diagram of the experimental procedure: platelets were allowed to interact with the cancer cells (cancer cell:platelet ratio = 1:50) for 3 hours. The interactions were stopped by extensively washing the

cancer cells with PBS-/-. the cancer cells were recorded to detect any changes in cancer cell behavior for another 3, 21 or 45 hours. **(B)** Representative images of three independent experiments of PANCO2 mock or PANCO2 low FN1 cells, in the presence (+) or in absence (−) of platelets over time up to 24 hours (holotomographic microscopy bars, 10 μm). **(C)** The curves, generated using the Cell Death Assay module, represent the percentage of cell death in PANCO2 mock (solid line) or PANCO2 + platelets (solid line with circle) cells **(upper panel)**, and in PANCO2 low FN1 (dashed line) or PANCO2 low FN1 + platelets (solid line with triangle) cells **(middle panel)** over a 24-hour period. The histogram in the **bottom panel** shows the mean area under the preceding curves (+/− SEM) from three independent experiments, with or without platelets (+/−) (ANOVA, ns = not significant). **(D)** Representative images **(left panel)** from nine independent wound healing assay experiments evaluating the migration of PANCO2 mock or PANCO2 low FN1 cells in the presence (+) or absence (−) of platelets over 24 hours (10X objective, scale bars = 250 μm). The graph **(right panel)** shows the mean percentage of area recovery (+/− SEM) from these nine independent experiments, reflecting the migration of PANCO2 mock or PANCO2 low FN1 cells, with (+) or without (−) platelets (5 images per experiment and per well, ANOVA test, *P < 0.0188). **(E)** Representative image **(left panel)** of FN1 detection by Western Blot in 15 μg of PANCO2 low FN1 lysates 6, 24 or 48 hours after their interaction with platelets (+) or not (−). Detection of GAPDH was used as a loading control for the experiment. Graph **(right panel)** shows the relative quantification of the FN1/GAPDH ratio in PANCO2 low FN1 cells normalized to PANCO2 low FN1 cells alone over time up to 48 hours. **(F)** Evaluation of cytokine expression in PANCO2 mock (light blue) or PANCO2 low FN1 (brown) cells after 6 **(left panel)** or 24 hours **(right panel)** interaction with platelets, normalized to PANCO2 or PANCO2 low FN1 cells alone, respectively, over a period of up to 24 hours using a multiplex ELISA workflow. Graphs represent the mean (+/− SEM) of relative fluorescence unit per cell normalized to cancer cells alone from three independent experiments.

allowed us to study the pancreas as a whole, since Capan-2 and PANC-1 come from the head of the pancreas, while BxPC-3 and MIA PaCa-2 come from the body and the tail [27]. Finally, cell lines such as PANC-1 and MIA PaCa-2 have a higher metastatic potential than HT29, BxPC-3 and Capan-2 [27,28]. This selection was designed to provide a comprehensive representation of digestive cancers, particularly in the absence of primary tumor samples from patients.

Initially, our focus was on the enzymes responsible for protein glycosylation, the glycosyltransferases (GT). In many cancers, several studies have shown that deregulation of GT expression leads to aberrant glycosylation with overexpression, deletion or truncation of specific carbohydrate structures, which can alter the behavior of cancer cells. Changes in glycosylation have even become a hallmark of cancer over the years [29,30]. Interestingly, Wandall *et al.* demonstrated in 2012 that platelets are capable of secreting or expressing functional GT and associated sugar nucleotides on their surface, enabling them to glycosylate extracellular molecules (proteins or lipids) [14]. We therefore hypothesized that platelets may be able to extracellularly glycosylate proteins present on the surface of cancer cells during their interaction, thereby modifying their phenotype.

Among the GTs highlighted in this process GALNT1, GALNT2 and GALNT3, members of the UDP-N-α-D galactosamine:polypeptide N-acetylgalactosaminyltransferases (GALNT) family, were described to be involved in different biological processes, including tumor progression, proliferation, and migration [31]. Surprisingly, in our model, we showed that the expression of GALNTs in platelets is very low and they are not functional. However, we demonstrated an increase in the total dry mass of cancer cells in the presence of platelets, reflecting an alteration in their composition. We observed variations in the expression of mRNAs encoding 43 different GT families; half of which were over-expressed (by 0 to 343%) and the other half underexpressed (by 0 to −25%) in platelet-educated cancer cells compared to cancer cells uneducated. Among the most significantly overexpressed are MGAT2, MGAT3 and B3GNT. The deregulation of these 3 GT families can have a significant impact on cancer progression. For example, MGAT2 (GnT-II) overexpression in neuroblastoma increases N-glycan structures promoting cell proliferation and invasion [32]. However, deregulation of MGAT3 (or GnT-III), as observed in breast cancer and melanoma, can inhibit the epithelial-mesenchymal transition (EMT) of cancer cells and thus suppress metastasis [33,34]. Finally, B3GNT3 is associated with immune cell infiltration in lung adenocarcinoma and, with pelvic lymph node metastasis in cervical cancer patients [35,36]. Thus, the education of cancer cells

by platelets in the tumor microenvironment may partially explain the increase in GT expression and this may potentiate their effect on cancer cells. To our knowledge, we have been the first to describe that the interaction of cancer cells with platelets induces a change in the cancer cell transcriptome itself.

We then extended this study to 124 additional genes encoding inflammatory and metastatic proteins and found that all were overexpressed (from 8% to 358%) in platelet-educated cancer cells compared to cancer cells alone. Among the most over-expressed genes were those known to be involved in metastasis, such as SERPINE1, MMP2 and 10, TIMP1 and FN1 [37–44]. Interaction between platelets and cancer cells increases the expression of SERPINE 1, an inhibitor of fibrinolysis. This over-expression, combined with the presence of microvesicles carrying active tissue factor, contributes to the pro-thrombotic profile of digestive cancers [6,45,46]. The presence of overexpression of RNA encoding tissue inhibitor of metalloproteinase 1 (TIMP1) seems to counterbalance the sharp increase in metalloproteases (MMP2 and 10) [47–49]. We therefore turned our attention to the effects of overexpressing FN1 on cancer cells. FN1 is a glycoprotein that plays a role in the adhesion and migration of cells and it has also been shown to promote tumor growth, metastasis, and chemoresistance in the tumor microenvironment [50]. Its ability to bind to integrins, such as the αvβ1 integrin, enables it to activate the Akt and Ras signaling pathways which promote cell proliferation [51]. Moreover, several studies demonstrated that FN1 can activates different signaling pathways such as focal adhesion kinase (FAK), SNAIL-related zinc-finger transcription factor (SLUG), signal transducer and activator of transcription 3 (STAT3), and ERK/MAP to trigger EMT and increase the metastatic potential of the cancer cells [50,52–55]. Finally, FN1 can, for example, activate the integrins α5β1 and α4β1, which increase the ability of cancer cells to transport drugs like doxorubicin out of the cytoplasm *via* the drug transporter ABCC1 [56]. This transport mechanism allows cancer cells to survive and grow even when exposed to high doses of chemotherapy [56]. Additionally, increased FN1 expression in pancreatic cancer cells makes them resistant to gemcitabine through activation of the ERK1/2 pathway, protecting the cells from its proapoptotic effects [57]. Here, we have shown that FN1 does not contribute to cell death, but that its overexpression in platelet-educated cancer cells reduces tumor cell migration. This suggests a potential anti-metastatic effect of FN1, which contradicts previous knowledge in this field. We also confirmed that platelets can stimulate FN1 transcription in cancer cells using cancer cells underexpressing FN1 by 2-fold. We have shown that the mechanism involved appears to be through cytokines such as EGF, PDGF, TNF-α, IFN-γ or IL-6, which activate signaling pathways within the cancer cells themselves that lead to FN1 production. These cytokines are all known to be involved in FN1 production though different signaling pathways such as MAPK/PI3K/Akt, STAT3 or JAK/STAT1 pathways [17–21].

Interestingly, FN1 glycosylation plays a role in the metastatic process. Studies have highlighted that in oral and mammary squamous cell carcinomas, FN1 undergoes a specific glycosylation that contributes to cell invasion and multidrug resistance of cancer cells [58,59]. This particular glycosylation, known as "oncofoetal FN1", is characterized by the addition of a glycan to a specific Thr in the C-terminal region of FN1 [58]. This is found exclusively in cancer and embryonic tissues [59]. A study showed that changes in the N-glycosylation of FN1 modulate integrin-mediated signaling to induce mediated cell adhesion and directed cell migration [60]. Given our previous findings on the variations in GT expression in platelet-educated cancer cells, it would be valuable to further investigate the glycosylation of FN1. Specifically, it would be interesting to examine whether there are changes in the carbohydrate motifs of FN1 after the interaction between cancer cells and platelets.

In addition to the increased expression of metastasis-related genes in platelet-educated cancer cells, our study also revealed the overexpression of tumor suppressor genes like

CTBP1, APC and TP53 [61–63]. While the pro-metastatic effect of overexpressed metastasis-related genes is clear, the impact of upregulation of tumor suppressor genes is more complex. Overexpression of CTBP1 has been observed in metastatic prostate cancer cells and leads to its mislocalization. As a result, this tumor-suppressing gene becomes pro-tumorigenic [61]. In contrast APC and TP53 mutations usually lead to their inactivation, affecting many cellular processes such as resistance to chemotherapy [62,63]. In our case, however, their unaltered forms are overexpressed, suggesting that they still may exert their tumor suppressor functions. These results highlight the dual role of platelets in the tumor microenvironment.

Finally, we have also showed the overexpression of genes whose role in cancer is not yet well understood such as IL2RG, CACNA1D and GNRH1. IL2RG encodes the γ subunit of the IL-2, IL-4, IL-7 and IL-21 receptors which plays crucial roles in the development and function of different immune cells such as NK, T and B cells. Knockdown of IL2RG is a common method to generate mouse models with impaired immune function, allowing the study of human immunodeficiency diseases. Although its direct contribution in cancer development is unclear, it has been found to be overexpressed in advanced gastric cancer and associated with poor survival rates [64]. Several studies have demonstrated that elevated levels of IL2RG as well as genes such as IL2RA, IL7R and IFNγ can promote melanoma metastasis by increasing intratumoral regulatory T cells through activation of the JAK-STAT signaling pathway [65]. In 2017, Ayars and al, using CRISPR-mediated knockout of *IL2RG* in orthotopically implanted pancreatic cancer cells, observed a reduction in tumor growth in mice and decreased JAK3 in orthotopic tumors. These findings suggest that the IL2Rγ/JAK3 signaling pathway may contribute to the growth of pancreatic cancer cells *in vivo* [66]. CACNA1D encodes a voltage-gated, L-type, α-subunit 1D calcium channel. It has been linked to adenomas (benign tumors) that produce aldosterone and endocrine disorders. McKerr *et al.* demonstrated that its modulation could induce a therapeutic advantage in prostate cancer, and several teams have also shown that its expression varies depending on the type of cancer, which means that it is referred to as a potential prognostic marker [67,68]. However, the mechanisms underlying its involvement in cancer are not well defined. Finally, studies have yielded mixed results regarding the association between GNRH1 and cancer. Some have found no clear association, while others have observed gender differences in bladder cancer prognosis or suggested that GNRH1 is a prognostic factor for the spread of tumor cells or cell cycle in breast and prostate cancers [69–71]. Taken together, our results shed light on interesting new target genes in platelet-educated cancer cells, to either inhibit or enhance, depending on their effects on cancer progression and metastasis.

In our study, we investigated the interactions between washed platelets and cancer cells *in vitro*. This approach allowed us to isolate and specifically study the effects of platelets on cancer cell gene expression without interference from other components such as plasma, and to establish a proof-of-concept for the education of cancer cells by platelets. This simplified model, however, does not fully account for the complexity of the tumor microenvironment or the dynamic nature of the blood compartment, which involve interactions among various cell types, the extracellular matrix, and soluble factors. To overcome these limitations, more sophisticated *in vitro* systems—such as 3D cell cultures, organ-on-chip models, or co-culture systems that better replicate tissue architecture—could offer deeper insights. Additionally, *in vivo* models, including patient-derived xenografts (PDX) and techniques like intravital microscopy, provide opportunities to observe platelet-cancer cell interactions in real-time within their native context. These alternative approaches represent promising next steps for exploring the biological significance of our findings in a more physiologically accurate setting.

In conclusion, we have demonstrated for the first time that platelets can educate cancer cells by inducing an increase or decrease in the transcription of genes related to glycosylation,

inflammation, and metastasis. Although the underlying mechanisms of this process require further investigation, it represents a promising approach for potential therapeutic interventions in cancer treatment.

## Supporting information

**S1 Fig. Representative images of three independent experiments of PANC-1 cells in the presence (+) or absence (−) of platelets over time up to 48 hours (holotomographic microscopy bars, 100 μm).** Segmentation was performed using "Cell Death Assay" module, excluding platelets from the analysis. Colors represent live cells (green) or dead cells (yellow).
(TIFF)

**S2 Fig. Representative images and corresponding quantification of glycan motif detection using PHA-L lectin by Western Blot of pancreatic cancer cell lysates (PANC-1, 40 μg) interacting (+) or not (−) with platelets for 3 hours.** GAPDH expression was detected as a loading control for the experiment. The graphs show the mean of integrated density of glycan motif expression normalized to cancer cells alone.
(TIF)

**S3 Fig. Representative images and corresponding quantification of MGAT2 by Western Blot of pancreatic cancer cell lysates (PANC-1, 40 μg) interacting (+) or not (−) with platelets for 3 hours.** GAPDH expression was detected as a loading control for the experiment. The graphs show the mean of integrated density of MGAT2 expression normalized to cancer cells alone.
(TIF)

**S1 File. Raw images.**
(PDF)

## Author contributions

**Formal analysis:** Mélanie Langiu, Lydie Crescence.

**Funding acquisition:** Diane Mège, Christophe Dubois.

**Investigation:** Mélanie Langiu, Lydie Crescence, Diane Mège, Christophe Dubois.

**Methodology:** Mélanie Langiu, Lydie Crescence.

**Supervision:** Laurence Panicot-Dubois.

**Validation:** Diane Mège, Laurence Panicot-Dubois, Christophe Dubois.

**Visualization:** Laurence Panicot-Dubois, Christophe Dubois.

**Writing – original draft:** Laurence Panicot-Dubois, Christophe Dubois.

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
