## [Decision Letter · Decision Letter 0]

29 Sep 2024

Dear Dr. Dubois,

Thank you for submitting your manuscript to PLOS ONE. After careful consideration, we feel that it has merit but does not fully meet PLOS ONE’s publication criteria as it currently stands. Therefore, we invite you to submit a revised version of the manuscript that addresses the points raised during the review process.

We look forward to receiving your revised manuscript.

Kind regards,

Li Yang, M.D.

Academic Editor

PLOS ONE

Journal Requirements:

2. To comply with PLOS ONE submissions requirements, in your Methods section, please provide additional information regarding the experiments involving animals and ensure you have included details on methods of sacrifice,  efforts to alleviate suffering.

4. Thank you for stating the following in your Competing Interests section: None 

6. We notice that your supplementary figures are uploaded with the file type 'Figure'. Please amend the file type to 'Supporting Information'. Please ensure that each Supporting Information file has a legend listed in the manuscript after the references list.

Reviewers' comments:

Reviewer's Responses to Questions

**Comments to the Author**

1. Is the manuscript technically sound, and do the data support the conclusions?

Reviewer #1: Yes

Reviewer #2: Yes

Reviewer #3: Yes

2. Has the statistical analysis been performed appropriately and rigorously?

Reviewer #1: Yes

Reviewer #2: I Don't Know

Reviewer #3: Yes

3. Have the authors made all data underlying the findings in their manuscript fully available?

Reviewer #1: Yes

Reviewer #2: Yes

Reviewer #3: Yes

4. Is the manuscript presented in an intelligible fashion and written in standard English?

Reviewer #1: Yes

Reviewer #2: Yes

Reviewer #3: Yes

Reviewer #1: The manuscript by Langiu et. al provides evidence of the platelets educating cancer cells in tumor microenvironment. The results are very interesting and the experiments are well designed.

I have following questions to the authors,

1) In the earlier manuscript (Cancer res, 2020), the authors show that the platelets in the tumor microenvironment inhibits cancer growth but increases metastasis. This is contradicting to what they observe in this manuscript where expression FN1 is upregulated there by slowing down the migration of cancer cells educated by platelets. How do they explain this phenomenon?

2) Authors go on to show that the MGAT2, GALNT3 genes are overexpressed in educated cancer cells. Was this increased gene expression correlate with the increased protein expression? Having a western blot for these proteins strengthens this claim.

3) the expression of VVL recognised glycans appear to be over expressed in PANC1 cells which is opposite to the claim from the authors. Was this the trend they saw with PANC1 cells if so, what might be the reason?

4) How do the authors explain the increased expression of TNF, IFN and IL-6 cytokines in platelet-educated cancer cells, which are pro-inflammatory and pro-apoptotic. Why is there no difference in the cell death?

5) According to the Cancer Res, 2020 manuscript from the authors, platelets-interaction with the cancer cells induces recruitment of macrophages and possible clearance of the cancer cells. Ref 12 in the manuscript also claim that platelets secrete alpha-granules that contains RANTES, MIF, IL-4. Can the change in the gene expression seen in cancer cells be a downstream event of the signaling pathways turned on by the cytokines? How does educating differs from cell-cell interaction?

6) It would be very interesting to see how these Plate-educated cancer cells behave in -vivo? Are these educated cancer cell more aggressive or less aggressive than normal cancer cells?

Finally, some of the sentences are too long and confusing. Kindly restructure them for easy read.

Reviewer #2: A manuscript entitled “Consequences of platelet-educated cancer cells on the expression of inflammatory and metastatic glycoproteins.” has been submitted for consideration to PlosOne.

The authors investigated the consequences of cancer cell education on proteins involved in glycosylation, inflammation and metastasis.

The authors found

- the interaction of cancer cells with platelets induced a change in the transcription of GT-encoding genes in the cancer cells themselves.

- 124 genes encoding for proteins associated with inflammation and metastasis were overexpressed in platelet-educated cancer cells, including fibronectin (FN1), a key glycoprotein involved in the adhesion processes.

The authors elegantly demonstrated: That platelets influence the composition of cancer cells, which may lead to changes in their behavior within the tumor microenvironment as well as the formation of metastasis.

This is a very interesting fundamental study, congratulation; the manuscript is well-written. The methodological aspect is very well-detailed and perfectly suited to the scientific hypothesis.

The minor Comments concerns:

Introduction:

- It is important to detail the notion of ‘educated’ in order to better define the original concept of the study.

- In addition to their role in haemostasis and thrombosis, platelets have an immune/inflammatory role that could be discussed in the introduction.

- Different cancer cell lines were used in this work. It would be interesting to define in the discussion the differences between these lines and the interest of diversifying the ‘cell culture’ approach to respond to the scientific hypothesis.

- The platelets in this study are washed, so it would be interesting in the discussion to challenge this point versus unwashed platelets by highlighting the choice of model and possible differences in terms of results that could have been observed with platelets closer to their physiological environment.

- It cannot be ruled out that during mRNA extraction from the cancer cell culture condition with platelets, some of the genetic material extracted is of platelet origin. This point needs to be addressed in the discussion.

- A paragraph on the limitations of the study should appear in the discussion.

- Data access for research purposes: “The experiments described in this manuscript using human platelets were performed after obtention of the IRB approval the 16th of May 2022.” you need to give your registration number

- The use of AI in the analysis is mentioned, but the description of this AI in the materials and methods section is not detailed enough.

Reviewer #3: The manuscript by Langiu et al. contributes to the group’s study of the role of platelets in cancer progression and metastasis. In this paper, the authors study the effect of platelet exposure on the expression glycosyltransferases and of genes involved in inflammation and metastasis. The paper is well-written, and the results are interesting and scientifically sound. I have a few suggestions to improve some parts of the manuscript that I feel need some clarification.

Materials and methods:

In this section, the methods and reagents used need to be better described:

1-Please indicate the source and catalog number of all primary and secondary antibodies used in all of the experiments (WB, Immunofluorescence, flow cytometry). Also indicate the dilutions and concentrations used for each antibody.

2- Please indicate the source and catalog number for the lectins and fluorescent conjugates used for protein glycosylation analysis. Please, also describe how this experiment was performed.

3-Please describe the experimental conditions used for the glycosyltransferase activity essay.

4-For the mRNA extraction please name the reagents used and indicate if mRNA was separated from other RNAs. If it was not, you did not extract mRNA, you extracted total RNA. Please make the necessary corrections. Please indicate if RNA integrity was accessed and by which method. UV ratios are not a good measure of RNA integrity.

5- For reverse-transcription, the authors state that 2 ug of mRNA were used. However, they did not describe the isolation of mRNA and their methods are not clearly described. Was total RNA or purified mRNA used for reverse-transcription? T

Results:

1-In figure 2A the authors have a column titled pancreatic cancer cells but in the legend they state that this is from BxPC-3 and Capan-2 cells. Could the authors please split this column in two and re-make the figure with a column for the BxPC-3, one for Capan-2 and one for platelets?

**Do you want your identity to be public for this peer review?** For information about this choice, including consent withdrawal, please see our Privacy Policy

Reviewer #1: No

Reviewer #2: No

Reviewer #3: **Yes: ** Rodolpho Mattos Albano

---

## [Author Response · Author response to Decision Letter 0]

3 Dec 2024

Responses to Reviewer Comments:

Reviewer #1: The manuscript by Langiu et al. provides evidence of the platelets educating cancer cells in tumor microenvironment. The results are very interesting, and the experiments are well designed. I have following questions to the authors,

1) In the earlier manuscript (Cancer res, 2020), the authors show that the platelets in the tumor microenvironment inhibits cancer growth but increases metastasis. This is contradicting to what they observe in this manuscript where expression FN1 is upregulated there by slowing down the migration of cancer cells educated by platelets. How do they explain this phenomenon?

Answer: We are grateful to the reviewer for this question. The two studies are not directly comparable as they involve different types of cancer—colorectal versus pancreatic. Additionally, our current study focuses solely on the effect of platelets on cancer cells without considering the microvesicular and microenvironmental compartments. The content of these microvesicles can vary between cancers, as can the response of cancer cells in the presence of platelets. It is also worth noting that the presence of fibronectin at the site of premetastatic niches may favor the development of metastasis (for a detailed review, please refer to Kaplan RN et al., Cancer Res. 2006). In fact, we are currently conducting a program on this specific point.

2) Authors go on to show that the MGAT2, GALNT3 genes are overexpressed in educated cancer cells. Was this increased gene expression correlate with the increased protein expression? Having a western blot for these proteins strengthens this claim.

Answer: We thank the reviewer for this remark. However, concerning the GALNT3, we did show a slit underexpression of GALNTs at the RNA level (figure 3A). We took the liberty of carrying out a western blot for an enzyme of the GALNTs family that is not expressed by platelets, GALNT3. We show a decrease in GALNT3 protein expression. We have not included this result in the manuscript, as we cannot link it to the GALNT family as a whole.

For the MAGT2 enzyme, overexpression of its mRNA is clearly observed in figure 3A. We performed western blots showing a tendency to increase the expression of MATG2 and sugar motifs created by MAGT2 (lectin PHA-L) after 3 hours of interaction. These western blots are shown in the supplementary figure S6 and S7.

LEGEND:

(S6) Representative images and corresponding quantification of glycan motif detection using PHA-L lectin by Western Blot of pancreatic cancer cell lysates (PANC-1, 40 µg) interacting (+) or not (-) with platelets for 3 hours. GAPDH expression was detected as a loading control for the experiment. The graphs show the mean of integrated density of glycan motif expression normalized to cancer cells alone.

(S7) Representative images and corresponding quantification of MGAT2 by Western Blot of pancreatic cancer cell lysates (PANC-1, 40 µg) interacting (+) or not (-) with platelets for 3 hours. GAPDH expression was detected as a loading control for the experiment. The graphs show the mean of integrated density of MGAT2 expression normalized to cancer cells alone.

We observed a trend towards an increase after only 3 hours of interaction with platelets. We have added these results as an additional figure and in the first part of the results as follows: “These findings appear to be validated at the protein level, showing a trend of increased expression of MGAT2 and MGAT2-catalyzed motifs recognized by the lectin PHA-L as early as 3 hours following the interaction between cancer cells and platelets (S6 and S7 Fig).”

3) The expression of VVL recognised glycans appear to be over expressed in PANC1 cells which is opposite to the claim from the authors. Was this the trend they saw with PANC1 cells if so, what might be the reason?

Answer: We thank the reviewer for this request. We have performed a new Western blot to target the carbohydrate motifs recognized by the VVL lectin. This membrane gives a more representative view of the results, although the quantification from the previous one also supports this result. We've replaced the previous Western blot with the new one (also in the corresponding graph), which changes the significance of the results (from **P=0.0090 to **P=0.0086).

4) How do the authors explain the increased expression of TNF, IFN and IL-6 cytokines in platelet-educated cancer cells, which are pro-inflammatory and pro-apoptotic. Why is there no difference in the cell death?

Answer: We thank the reviewer for this question. As described in the main manuscript, we examined the expression of 21 cytokines in cancer cell lysates that had either interacted with platelets or not. We specifically focused on TNFα, IFNγ, IL-6, EGF, and PDGF-BB, as these five cytokines are known to influence FN1 production. Additionally, as shown in the figure below, there is a concomitant increase in the expression of GM-CSF, IL-2, and IL-15, which are known for their anti-apoptotic effects under certain conditions. This may explain the lack of difference in cell death between the two conditions, as the combined effects of these cytokines are likely to compensate for each other, bearing in mind that over 21 cytokines are present in a cell.

To avoid confusion, we add these sentences in the main text: “It should be noted that TNF-α, IL-6 and IFNγ are also known to be pro-apoptotic [22–24]. However, other anti-apoptotic molecules such as GM-CSF, IL-2, and IL-15 are also over-expressed and may counterbalance this effect, leading to no significant difference in cell death between the two lines tested over the 24-hour period.”

Figure: Evaluation of cytokine expression in PANCO2 mock (light blue) or PANCO2 low FN1 (brown) cells after 24 hours interaction with platelets, normalized to PANCO2 or PANCO2 low FN1 cells alone, respectively, over a period of up to 24 hours using a multiplex ELISA workflow. Graphs represent the mean (+/- SEM) of relative fluorescence unit per cell normalized to cancer cells alone from three independent experiments.

5) According to the Cancer Res, 2020 manuscript from the authors, platelets-interaction with the cancer cells induces recruitment of macrophages and possible clearance of the cancer cells. Ref 12 in the manuscript also claim that platelets secrete alpha-granules that contains RANTES, MIF, IL-4. Can the change in the gene expression seen in cancer cells be a downstream event of the signaling pathways turned on by the cytokines? How does educating differs from cell-cell interaction?

Answer: We thank the reviewer for this comment. The reviewer is right. However, the exact mechanism driving the change in gene expression in platelet-educated cancer cells remains unknown. Further experiments would be needed to clarify this, for example using knockouts targeting cytokine signaling pathways and related mechanisms. The education of cancer cells by platelets occurs through both direct and indirect processes. The difference between these pathways is that the direct pathway involves the transfer of material from platelets to cancer cells. To determine the specific contribution of each pathway, transwell assays should be used, which would allow the cancer cells to be physically separated from the platelets. This approach would help to assess whether the effect on the transcriptome is primarily due to direct interaction or to indirect factors such as cytokines, microparticles (MPs) and other soluble mediators. These are potential future directions for this line of research.

6) It would be very interesting to see how these Plate-educated cancer cells behave in vivo? Are these educated cancer cell more aggressive or less aggressive than normal cancer cells?

Answer: We thank the reviewer for this remark. The observation that platelet-educated cancer cells exhibit increased fibronectin expression, enhancing their adhesive properties, could have contrasting implications depending on the context. In the primary tumor, this increased adhesion may act as an anti-metastatic factor, whereas in the bloodstream, it could promote metastasis. Therefore, investigating their behavior in vivo, both in the bloodstream and within the primary tumor, would be a valuable direction for future research. However, it is important to note that this current study serves as a proof of concept and is based solely on in vitro experiments. The in vivo effects will be addressed in a subsequent study.

Finally, some of the sentences are too long and confusing. Kindly restructure them for easy read.

Thank you for your comments. We've done a complete proofreading of the article and removed some sentences that may have been too long.

Reviewer #2: A manuscript entitled “Consequences of platelet-educated cancer cells on the expression of inflammatory and metastatic glycoproteins.” has been submitted for consideration to PlosOne.

The authors investigated the consequences of cancer cell education on proteins involved in glycosylation, inflammation, and metastasis. The authors found:

• The interaction of cancer cells with platelets induced a change in the transcription of GT-encoding genes in the cancer cells themselves.

• 124 genes encoding for proteins associated with inflammation and metastasis were overexpressed in platelet-educated cancer cells, including fibronectin (FN1), a key glycoprotein involved in the adhesion processes.

The authors elegantly demonstrated: That platelets influence the composition of cancer cells, which may lead to changes in their behavior within the tumor microenvironment as well as the formation of metastasis.

This is a very interesting fundamental study, congratulation; the manuscript is well-written. The methodological aspect is very well-detailed and perfectly suited to the scientific hypothesis.

The minor Comments concerns:

Introduction:

1) It is important to detail the notion of ‘educated’ to better define the original concept of the study.

Answer: We fully agree with the reviewer and have included an extended definition of 'education' in the introduction (lines 22 to 28). Please find here the text: “Interestingly, these studies also highlighted the opposite concept of plated-educated cancer cells. Indeed, platelets can modulate cancer cell behavior through both direct and indirect mechanisms. Direct interactions involve the transfer of platelet biomolecules such as lipids, proteins, and RNA from platelets to cancer cells. In contrast, the indirect pathway occurs when the interaction with platelets induces the activation of signaling pathways in cancer cells themselves, leading to a change in their molecular composition and, ultimately, in their behavior.”

2) In addition to their role in haemostasis and thrombosis, platelets have an immune/inflammatory role that could be discussed in the introduction.

Answer: We agree with the reviewer that platelets play a role in immunity and inflammation. However, we believe that adding a paragraph on this topic would disrupt the flow of our introduction.

3) Different cancer cell lines were used in this work. It would be interesting to define in the discussion the differences between these lines and the interest of diversifying the ‘cell culture’ approach to respond to the scientific hypothesis.

Answer: We thank the reviewer for this comment. In the discussion section, we have added a justification for our choice of the various cell lines, further highlighting the robustness of ours results: To this end, we conducted an in vitro study using cell lines derived from human digestive cancers such as colorectal adenocarcinoma (HT29), pancreatic adenocarcinoma (BxPC-3, Capan-2), and pancreatic carcinoma (PANC-1, MIA PaCa-2). Colorectal cancer, associated with a moderate risk of thrombosis, and pancreatic cancer, linked to a higher thrombotic risk, were selected to explore the interaction between cancer cells and platelets. We chose to focus on pancreatic cancer because of its higher mortality rate. To study this interaction in detail, we used four pancreatic cancer cell lines: BxPC-3, Capan-2, PANC-1 and MIA PaCa-2. These cell lines were chosen for their diversity in terms of gender, age, and genetic mutations [24]. All except BxPC-3 (which lacks the Kras mutation) carry the four most common mutations in pancreatic cancer: Kras, TP53, CDKN2A/p16 and SMAD4-DPC4 [24,25]. Notably, HT29 also carries a TP53 mutation, which is common in colorectal cancers [25]. In addition, these cell lines allowed us to study the pancreas as a whole, since Capan-2 and PANC-1 come from the head of the pancreas, while BxPC-3 and MIA PaCa-2 come from the body and the tail [24]. Finally, cell lines such as PANC-1 and MIA PaCa-2 have a higher metastatic potential than HT29, BxPC-3 and Capan-2 [24,25]. This selection was designed to provide a comprehensive representation of digestive cancers, particularly in the absence of primary tumor samples from patients.

4) The platelets in this study are washed, so it would be interesting in the discussion to challenge this point versus unwashed platelets by highlighting the choice of model and possible differences in terms of results that could have been observed with platelets closer to their physiological environment.

Answer: We'd like to thank the Reviewer for pointing this out, so we've added the following paragraph: “In our study, we investigated the interactions between washed platelets and cancer cells in vitro. This approach allowed us to isolate and specifically study the effects of platelets on cancer cell gene expression without interference from other components such as plasma, and to establish a proof-of-concept for the education of cancer cells by platelets. This simplified model, however, does not fully account for the complexity of the tumor microenvironment or the dynamic nature of the blood compartment, which involve interactions among various cell types, the extracellular matrix, and soluble factors.”

5) It cannot be ruled out that during mRNA extraction from the cancer cell culture condition with platelets, some of the genetic material extracted is of platelet origin. This point needs to be addressed in the discussion.

Answer: We thank the reviewer for this feedback. Following our 3 hours cancer cell-platelet interactions, we perform five washes with PBS to remove the platelets and analyze only the cancer cells in all our experiments. To convince the reviewer of the absence of free platelets and therefore of platelet-derived mRNA in our reaction medium, we counted the platelet marker CD41 on the cancer cells as a function of the number of washes. Note that we performed all our experiments with 5 washes. We can therefore consider that the platelets detected are only those interacting with our cancer cells, as shown by our additional holotomographic microscopy data. What's more, platelets contain a limited quantity of mRNA (1 mg of RNA requires 5,004,787,086 platelets and 28,735,632 cancer cells, 174 times less). We can therefore practically rule out a significant quantitative contribution of RNA from platelets.

*SN = supernatant

6) A paragraph on the limitations of the study should appear in the discussion.

Answer: We are in full agreement with the reviewer and have therefore added a paragraph on the limitations of the study at the end of the discussion, in accordance with the reviewer's recommendations: “To overcome these limitations, more sophisticated in vitro systems—such as 3D cell cultures, organ-on-chip models, or co-culture systems that better replicate tissue architecture—could offer deeper insights. Additionally, in vivo models, including patient-derived xenografts (PDX) and techniques like intravital microscopy, provide opportunities to observe platelet-cancer cell interactions in real-time within their native context. These alternative approaches represent promising next steps for exploring the biological significance of our findings in a more physiologically accurate setting.”

7) Data access for research purposes: “The experiments described in this manuscript using human platelets were performed after obtention of the IRB approval the 16th of May 2022.” you need to give your registration number

Answer: We thank the

---

## [Decision Letter · Decision Letter 1]

15 Dec 2024

Dear Dr. Dubois,

Thank you for submitting your manuscript to PLOS ONE. After careful consideration, we feel that it has merit but does not fully meet PLOS ONE’s publication criteria as it currently stands. Therefore, we invite you to submit a revised version of the manuscript that addresses the points raised during the review process.

We look forward to receiving your revised manuscript.

Kind regards,

Li Yang, M.D.

Academic Editor

PLOS ONE

Journal Requirements:

**Additional Editor Comments:**

I am pleased to report that this paper has been improved a lot and all the reviewers have approved the publication of your manuscript. However, before I can recommend the final editorial decision to our journal office, some minor issues need your attention. Please further address my concerns. Thanks for the chance to assess your work.

1) Are there any ongoing or completed clinical trials related to your study topic? If yes, please consider to add a table to summarize these studies.

2) I note that there are many Figures (6 figures and 9 supplemental figures), which occupy large room for publication. The image composition and arrangement are really unaesthetic. Actually, each Figure contains really little data, making each little figure A/B/C/D... within a big Figure relatively large, especially for Figure 1, 2, 3, 4. .. Please consider to merge some of the figures into one and make each Figure looks compact. The Figure 5 and Figure 6 may be better for reference.

3) Some supplemental figures are important to clarify your core findings as well. Please consider to move them into the main figures, such as S2, S4, S5, S8....

4) In a word, authors should comprehensively rearrange all the figures and supplemental figures and make them meet the high standard for final publication.

5) I encourage authors should attach all the raw data in a supplemental material.

Reviewers' comments:

Reviewer's Responses to Questions

**Comments to the Author**

Reviewer #1: All comments have been addressed

Reviewer #2: All comments have been addressed

Reviewer #3: All comments have been addressed

2. Is the manuscript technically sound, and do the data support the conclusions?

Reviewer #1: Yes

Reviewer #2: Yes

Reviewer #3: Yes

3. Has the statistical analysis been performed appropriately and rigorously?

Reviewer #1: Yes

Reviewer #2: Yes

Reviewer #3: Yes

4. Have the authors made all data underlying the findings in their manuscript fully available?

Reviewer #1: Yes

Reviewer #2: Yes

Reviewer #3: Yes

5. Is the manuscript presented in an intelligible fashion and written in standard English?

Reviewer #1: Yes

Reviewer #2: Yes

Reviewer #3: Yes

Reviewer #1: the authors have satisfactorily answered all my concerns. I recommend the manuscript for publication.

Reviewer #2: I thank you very much for your consideration of my comments and I congratulate you on this very interesting study.

Reviewer #3: (No Response)

**Do you want your identity to be public for this peer review?** For information about this choice, including consent withdrawal, please see our Privacy Policy

Reviewer #1: No

Reviewer #2: No

Reviewer #3: **Yes: ** Rodolpho Mattos Albano

---

## [Author Response · Author response to Decision Letter 1]

19 Dec 2024

Responses to the Editor's comments

I am pleased to report that this paper has been improved a lot and all the reviewers have approved the publication of your manuscript. However, before I can recommend the final editorial decision to our journal office, some minor issues need your attention. Please further address my concerns. Thanks for the chance to assess your work.

1) Are there any ongoing or completed clinical trials related to your study topic? If yes, please consider to add a table to summarize these studies.

Answer: We thank the editor for this comment. We are not aware of any clinical trial related to our study.

2) I note that there are many Figures (6 figures and 9 supplemental figures), which occupy large room for publication. The image composition and arrangement are really unaesthetic. Actually, each Figure contains really little data, making each little figure A/B/C/D... within a big Figure relatively large, especially for Figure 1, 2, 3, 4. .. Please consider to merge some of the figures into one and make each Figure looks compact. The Figure 5 and Figure 6 may be better for reference.

Answer: We have reduced the number of figures from 6 to 4.

3) Some supplemental figures are important to clarify your core findings as well. Please consider to move them into the main figures, such as S2, S4, S5, S8....

Answer: We have included previous additional figures in the main figures.

4) In a word, authors should comprehensively rearrange all the figures and supplemental figures and make them meet the high standard for final publication.

Answer: We have modified our figures following the editor's advices.

5) I encourage authors should attach all the raw data in a supplemental material.

Answer: we'll gladly give all the raw data to anyone who asks us for them.

---

## [Editor Report · Decision Letter 2]

22 Dec 2024

Consequences of platelet-educated cancer cells on the expression of inflammatory and metastatic glycoproteins

PONE-D-24-21387R2

Dear Dr. Dubois,

We’re pleased to inform you that your manuscript has been judged scientifically suitable for publication and will be formally accepted for publication once it meets all outstanding technical requirements.

Kind regards,

Li Yang, M.D.

Academic Editor

PLOS ONE

Additional Editor Comments (optional):

Thanks for your efforts to address my concerns. I am happy to report that this paper can be accepted in its current form. You will be notified the official decision letter of acceptance when our editorial team complete all the technical inspection.
---

## [Editor Report · Acceptance letter]

PONE-D-24-21387R2

PLOS ONE

Dear Dr. Dubois,

I'm pleased to inform you that your manuscript has been deemed suitable for publication in PLOS ONE. Congratulations! Your manuscript is now being handed over to our production team.

Kind regards,

on behalf of

Dr. Li Yang

Academic Editor

PLOS ONE